



# Do small and large floods have the same drivers of change? A regional attribution analysis in Europe

Miriam Bertola[1], Alberto Viglione[2], Sergiy Vorogushyn[3], David Lun[1], Bruno Merz[3,4], and
Günter Blöschl[1]

[1]Institute of Hydraulic Engineering and Water Resources Management, Vienna University of Technology, Karlsplatz 13, 1040
Vienna, Austria
[2]Department of Environment, Land and Infrastructure Engineering (DIATI), Polytechnic University of Turin, Corso Duca
degli Abruzzi 24, 10129 Turin, Italy
[3]GFZ German Research Centre for Geosciences, Hydrology section, Telegrafenberg, 14473 Potsdam, Germany
[4]Institute for Environmental Sciences and Geography, University of Potsdam, Karl-Liebknecht-Straße 24-25, 14476 Potsdam,
Germany

**Correspondence:** Miriam Bertola (bertola@hydro.tuwien.ac.at)

**Abstract.** Recent studies have shown evidence of increasing and decreasing trends in mean annual floods and flood quantiles across Europe. Studies attributing observed changes in flood peaks to their drivers have mostly focused on mean annual floods. This paper proposes a new framework for attributing flood changes to potential drivers, as a function of return period (T), in a regional context. We assume flood peaks to follow a non-stationary regional Gumbel distribution, where the median flood and

the 100-year growth factor are used as parameters. They are allowed to vary in time and between catchments as a function of the drivers quantified by covariates. The elasticities of floods with respect to the drivers and the contributions of the drivers to flood changes are estimated by Bayesian inference. The prior distributions of the elasticities of flood quantiles to the drivers are estimated by hydrological reasoning and from the literature. The attribution model is applied to European flood and covariate data and aims at attributing the observed flood trend patterns to specific drivers for different return periods. We analyse flood

discharge records from 2370 hydrometric stations in Europe over the period 1960-2010. Extreme precipitation, antecedent soil moisture and snowmelt are the potential drivers of flood change considered in this study. Results show that, in northwestern Europe, extreme precipitation mainly contributes to changes in both the median ($q_2$) and 100-year flood ($q_{100}$), while the contributions of antecedent soil moisture are of secondary importance. In southern Europe, both antecedent soil moisture and extreme precipitation contribute to flood changes, and their relative importance depends on the return period. Antecedent soil

moisture is the main contributor to changes in $q_2$, while the contributions of the two drivers to changes in larger floods (T>10 years) are comparable. In eastern Europe, snowmelt drives changes in both $q_2$ and $q_{100}$.

## 1   Introduction

There is widespread concern that river flooding has become more frequent and severe during the last decades, and that human-induced climate change and other drivers will further increase flood discharge and damage in many parts of the world (IPCC,

2012; Hirabayashi et al., 2013). This concern has given rise to a large number of studies investigating past changes in flood





hazard, i.e. changes related to flood discharge, and flood risk, i.e. related to damage. The global pattern of increasing flood damage has been mainly attributed to increasing population, economic activities and assets in flood-prone areas (Bouwer, 2011; IPCC, 2012; Visser et al., 2014). In terms of changes in flood discharge, a variety of changes has been found (for shift in timing and trends in the magnitude of European floods, see Blöschl et al., 2017, 2019), and attempts to attribute detected

changes have not resulted in a clear picture about the contribution of the underlying drivers (for a review on detecting and attributing flood hazard changes in Europe see Hall et al., 2014).

The large majority of studies on past changes in flood hazard analysed the mean flood behaviour, using, for instance, the Mann-Kendall test to detect gradual changes or the Pettitt test for step changes in the mean or median annual flood (e.g. Petrow and Merz, 2009; Villarini et al., 2011; Mediero et al., 2014; Mangini et al., 2018). This focus may be misleading, since changes

in large floods may differ from those in the average behaviour. An illustrative example is the Mekong River, where studies found negative trends in the mean flood discharge, whereas the public perception suggested that the frequency of damaging floods had increased in the past decades. Delgado et al. (2009) resolved this mismatch by analysing the temporal change in flood discharge variability. They found an upward trend in interannual variability which outweighed the decreasing mean behaviour leading to contrasting trends in the mean flood and rare floods. This change in flood variability could be attributed

to changes in the Western Pacific monsoon (Delgado et al., 2012). Another recent example is the large-scale study of Bertola et al. (2020) which compared trends of small with those of large floods (i.e. the 2-year and the 100-year flood) across Europe. They found distinctive patterns of flood change which depend on the return period and catchment scale.

It has been widely acknowledged that drivers can differently affect small and large floods (e.g. Hall et al., 2014) and yet the focus has been mainly on changes in the mean flood behaviour. One reason for this may be the ability of quantifying

changes in the mean more robustly than those of larger floods. However, both from theoretical and practical perspectives, detection and attribution of flood changes as a function of the return period are of considerable interest for understanding how the non-linearity in the hydrological system plays out and for providing guidance for flood risk management. The shape of the flood frequency curve and its changes in time are a reflection of the interplay between atmospheric processes and catchment state (soil moisture and snow), with different characteristics depending on the region, climate and runoff generation processes

(Blöschl et al., 2013).

Rainfall itself may increase at different rates for small and extreme events in a changing climate. These changes may strongly differ depending on the region and season. In addition, changes in rainfall may be translated in a non-linear way into changes of various flood magnitudes due to the non-linearity of the catchment response. For example, Rogger et al. (2012) detected a change in the slope of the flood frequency curve and linked it to the interplay of catchment saturation and rainfall. Several

studies indicated changes in precipitation amounts/intensities for different rainfall quantiles that might translate into different changes of small and large floods. For Germany, Murawski et al. (2016) found an increasing variability of precipitation along with increasing mean in seasons other than summer, which leads to a disproportional increase of heavy precipitation. Van den Besselaar et al. (2013) detected a decrease of the return period of extreme precipitation (5, 10 and 20 years) over Europe in the past 60 years between 2 and 58%. Berg et al. (2013) found a disproportional increase of high-intensity, convective precipitation

with increasing temperature that goes beyond the Clasius-Clapeyron rate (7% per degree of temperature increase) compared





to low-intensity, stratiform precipitation. The review of a number of regional studies on past precipitation trends in Europe by Madsen et al. (2014) suggested a tendency for increasing extreme rainfalls. This trend seemed not to translate directly into positive trends in observed streamflow over large scales in Europe (Madsen et al., 2014). Similarly, Hodgkins et al. (2017) suggested that occurrence of floods with return periods of 25 to 100 years is dominated by multi-decadal climate variability

rather than by long-term trends based on the analysis of more than 1200 gauges in Europe and North America. The study suggested that occurrence rate of larger floods (50 and 100 years) increased slightly stronger compared to smaller floods (25 years) in Europe over the past about 50 years.

It has been observed that increases in precipitation extremes often do not translate in increasing floods (Madsen et al., 2014; Sharma et al., 2018). This is attributable to other factors which modulates flood response, such as initial soil moisture. For

example, Tramblay et al. (2013) found that, despite the increase in extreme precipitation, the fewer detected annual occurrences of extreme floods in 171 Mediterranean basins were likely caused by decreasing soil moisture. The relationship between the flow rate and the initial saturation state of the soil is often non-linear and the effect of antecedent soil moisture strongly depends on soil type and geology. The sensitivity of floods to initial soil moisture depends on flood magnitude, and runoff generation is more influential for smaller events. Vieux et al. (2009) analysed several watersheds in the Korean peninsula with a distributed

hydrologic model and found that the sensitivity of the watershed response to the initial degree of saturation is dependent on event magnitude. Zhu et al. (2018) simulated peak discharges for return periods of 2 to 500 years for several sub-watersheds in Turkey River in the Midwestern United States and found that antecedent soil moisture modulates the role of rainfall structure in simulated flood response, particularly for smaller events. Grillakis et al. (2016) analysed flash flood events in two Greek and one Austrian catchments, and found higher sensitivity of the smallest flood events to initial soil moisture, compared to

larger events. These results are consistent throughout the different regions and climates, confirming that the effects of initial soil moisture on flood response depend on flood magnitude.

Snow storage and melt are other important factors that modulate flood response in temperate and cold regions. Snowmelt represents the dominant flood generating process in northeastern Europe and rain-on-snow is relevant for regions in central and northwestern Europe (Berghuijs et al., 2019; Kemter et al., 2020). It was observed that in catchments where snowmelt

and rain-on-snow are the dominant flood generating processes, the shape of the flood frequency curve is likely to flatten out at large return periods due to the upper limit of energy available for melt (Merz and Blöschl, 2003; Merz and Blöschl, 2008). Reduction in spring and summer snow cover extents have been detected as a result of increasing spring temperature in the Northern Hemisphere (Estilow et al., 2015). Several studies in regions dominated by snowmelt-induced peak flows reported decrease in extreme streamflow and earlier spring snowmelt peak flows, likely caused by increasing temperature (Madsen et al.,

2014). The effects of changing snow storage and melt on the flood frequency curves likely depend on flood regimes and mixing of different flood generating processes in the catchments. For example, in Carinthia, in the very south of Austria, the major floods tend to occur in autumn, and spring snowmelt floods represent a smaller fraction of events with small magnitude (Merz and Blöschl, 2003). Hence, changes in snow cover and melt are expected to mainly affect the smaller floods in these climates. In contrast, in northeastern Europe where snowmelt is the dominant flood generating process of both small and large floods,

the effects of decreasing snowmelt are likely important for the entire flood frequency curve.





Overall, the contributions of different drivers to flood changes as a function of return period are currently not well understood. This is partly due to detection and attribution studies focusing generally on the mean annual flood. Several studies applied non-stationary frequency analysis to attribute past flood changes to potential drivers. These studies typically allowed the parameters of the probability distribution of floods to vary in time, using time-varying climatic covariates (e.g. Prosdocimi et al., 2014;

Šraj et al., 2016; Steirou et al., 2019) and, more rarely, catchment and river covariates (e.g. López and Francés, 2013; Silva et al., 2017; Bertola et al., 2019). They attempted to identify and select covariates in the non-stationary model that provide a better fit to the flood data than the alternative stationary model. However, these studies still aimed at attributing changes in the mean annual flood and did not separate the effects of drivers on floods associated with different return periods.

The aim of this paper is to address two science questions: (a) Is it possible to identify the relative contributions of different

drivers to observed flood changes across Europe as a function of the return period, and if so, (b) what is the magnitude and sign of these contributions across Europe? Regarding the first question, one possible outcome is for the data to provide evidence that the relative contributions differ, or alternatively, the data may contain insufficient information to separate the effects by return period. Regarding the second question, the interest resides in understanding the relative importance of potential drivers as a function of return period, provided that such information can be inferred from the data. In this study, we adopt a non-stationary

flood frequency approach to attribute observed flood changes to potential drivers, used as covariates of the parameters of the regional probability distribution of floods. Extreme precipitation, antecedent soil moisture and snowmelt are the potential drivers considered. The relative contribution of the different drivers to flood changes is quantified through the elasticity of flood quantiles with respect to each driver.

## 2 Methods

### 2.1 Regional driver-informed model

In this study, we use non-stationary flood frequency analysis to attribute observed flood changes across Europe (see e.g., Blöschl et al., 2019; Bertola et al., 2020) to potential drivers, used as time-varying covariates. In the spirit of Bertola et al. (2020), we formulate the flood model as a regional Gumbel model. The Gumbel distribution has two parameters (i.e. the location $\mu$ and scale $\sigma$ parameters) and its cumulative distribution function is:

$$F_X(x) = p = e^{-e^{-\frac{x-\xi}{\sigma}}} \tag{1}$$

The two Gumbel parameters can be inferred from knowledge of two flood quantiles, e.g., the 2-year and the 100-year flood. We adopt here the same alternative parameters as in Bertola et al. (2020), i.e. the 2-year flood $q_2$ and the 100-year growth factor $x'_{100}$. The T-year flood can be obtained with the following relationship:

$$q_T = q_2 \left(1 + a_T x'_{100}\right) \tag{2}$$

where $a_T = (y_T - y_2)/(y_{100} - y_2)$, with $y$ being the Gumbel reduced variate, which is related to the return period by:

$$y_T = -\ln\left(-\ln\left(1 - \frac{1}{T}\right)\right) = -\ln\left(-\ln p\right) \tag{3}$$





We adopt the following regional change model accounting for catchment area:

$$\ln q_2 = \ln \alpha_{2_0} + \gamma_{2_0} \ln S + \alpha_{2_1} \ln X_1 + \alpha_{2_2} \ln X_2 + \alpha_{2_3} \ln X_3 + \varepsilon \tag{4a}$$

$$\ln x'_{100} = \ln \alpha_{g_0} + \gamma_{g_0} \ln S + \alpha_{g_1} \ln X_1 + \alpha_{g_2} \ln X_2 + \alpha_{g_3} \ln X_3 \tag{4b}$$

$\quad \varepsilon \sim \mathcal{N}(0, \sigma)$

Where $X_1$, $X_2$ and $X_3$ are three covariates (i.e. time series of the potential drivers of flood change), $S$ is catchment area and the Greek symbols represent the parameters of the model to be estimated. The $\varepsilon$ term, here assumed normally distributed, accounts for additional local variability (i.e. not explained by catchment area and the covariates) of $q_2$.

The elasticity of the generic flood quantile $q_T$ with respect to the covariate $X_i$ is defined as:

$\quad$
$$S_{T,X_i} = \frac{X_i}{q_T} \frac{\partial q_T}{\partial X_i} = \alpha_{2_i} + \alpha_{g_i} \left( 1 - \frac{1}{1 + a_T x'_{100}} \right) \tag{5}$$

It represents the percentage change in $q_T$, due to a 1% change in $X_i$, i.e., how sensitive flood peaks are to changes in the drivers. However, the elasticity alone does not tell how much the flood quantiles have actually changed (in time) due to observed changes of the drivers. Hence, we define the contribution of $X_i$ to the changes in $q_T$ as:

$$C_{T,X_i} = \frac{X_i}{q_T} \frac{\partial q_T}{\partial X_i} \cdot \frac{1}{X_i} \frac{dX_i}{dt} \tag{6}$$

It represents the percentage change in $q_T$, due the actual change in $X_i$. The total change in $q_T$ due to the changes in the drivers, assuming that the contributions are additive, is:

$$\frac{1}{q_T} \frac{dq_T}{dt} = \sum_i C_{T,X_i} = \sum_i \frac{X_i}{q_T} \frac{\partial q_T}{\partial X_i} \cdot \frac{1}{X_i} \frac{dX_i}{dt} \tag{7}$$

A measure of relative contribution of $X_i$ to the change in $q_T$ is expressed here by:

$$R_{T,X_i} = \frac{abs(C_{T,X_i})}{\sum_i abs(C_{T,X_i})} \tag{8}$$

where $\sum_i R_{T,X_i} = 1$

In the change model, the flood and covariate data are pooled and used simultaneously to attribute any observed changes in floods to their drivers. This pooling increases the robustness of the estimates (see e.g., Viglione et al., 2016) but requires an assumption of homogeneity. Specifically, we assume here that for a given return period and catchment scale, the elasticities of the flood discharges to their drivers are uniform within the region. We do allow the drivers to vary between catchments.

We frame the estimation problem in Bayesian terms through a Markov chain Monte Carlo (MCMC) approach, using the R package rStan (Carpenter et al., 2017) which makes use of a Hamiltonian Monte Carlo algorithm to sample the posterior distribution (Stan Development Team, 2018). For each inference, we generate four chains of 10 000 simulations each with different initial values and we check for their convergence. We use prior information on the model parameters to constrain their estimation to hydrologically plausible values (see Sect. 2.5).





## 2.2 Spatial correlation of floods

Spatial correlation of floods is not directly accounted for in the proposed regional change model of Sect. 2.1 and it may result in underestimated sample uncertainties (see e.g., Stedinger, 1983; Castellarin et al., 2008; Sun et al., 2014). Here, we adopt an approach proposed by Ribatet et al. (2012) and based on the work of Smith (1990), consisting in a magnitude adjustment to the likelihood function in a Bayesian framework, which accounts for the overall dependence in space and allows to obtain reliable credible intervals. The adjusted likelihood is defined as:

$$L^*\left(\theta,\mathbf{y}\right) = L\left(\theta,\mathbf{y}\right)^k \tag{9}$$

where $L$ is the likelihood under the assumption of spatial independence, $\theta$ is the vector of unknown parameters and $k$ is the magnitude adjustment factor to be estimated, such as $0 < k \leq 1$ (see Appendix A). The magnitude adjustment factor $k$ represents the overall reduction of hydrological information in the data caused by the presence of spatial correlation and results in an inflated posterior variance of the parameters. If floods at different sites are spatially independent, $k$ is 1; on the contrary, if floods are strongly cross-correlated, k assumes values close to 0. In this latter case, the sample uncertainty resulting from the adjusted likelihood will be larger, compared to the model where spatial cross-correlation is not accounted for. For further details on the adjustment to the likelihood and its application to hydrological data see Smith (1990), Ribatet et al. (2012) and Sharkey and Winter (2019).

## 2.3 Data

Consistently with Blöschl et al. (2019) and Bertola et al. (2020), we analyse long series of annual maximum discharges between 1960 and 2010, from 2370 hydrometric stations in 33 European countries (https://github.com/tuwhydro/europe_floods). Stations affected by strong artificial alterations (such as large reservoirs in the proximity of the gauges) are not included in this database (Blöschl et al., 2019). The location of the stations is shown in Fig. 1. Their contributing catchment areas range from 5 to 100 000 km$^2$ and the median record length is 51 years. The catchment boundaries relative to each hydrometric station are derived from the CCM River and Catchment Database (Vogt et al., 2007). Daily gridded precipitation and mean surface temperature is obtained from the E-OBS dataset (version 18.0e, resolution 0.1 deg; Cornes et al. (2018)). It covers the area 25N-71.5N x 25W-45E for the period 1950-2018.

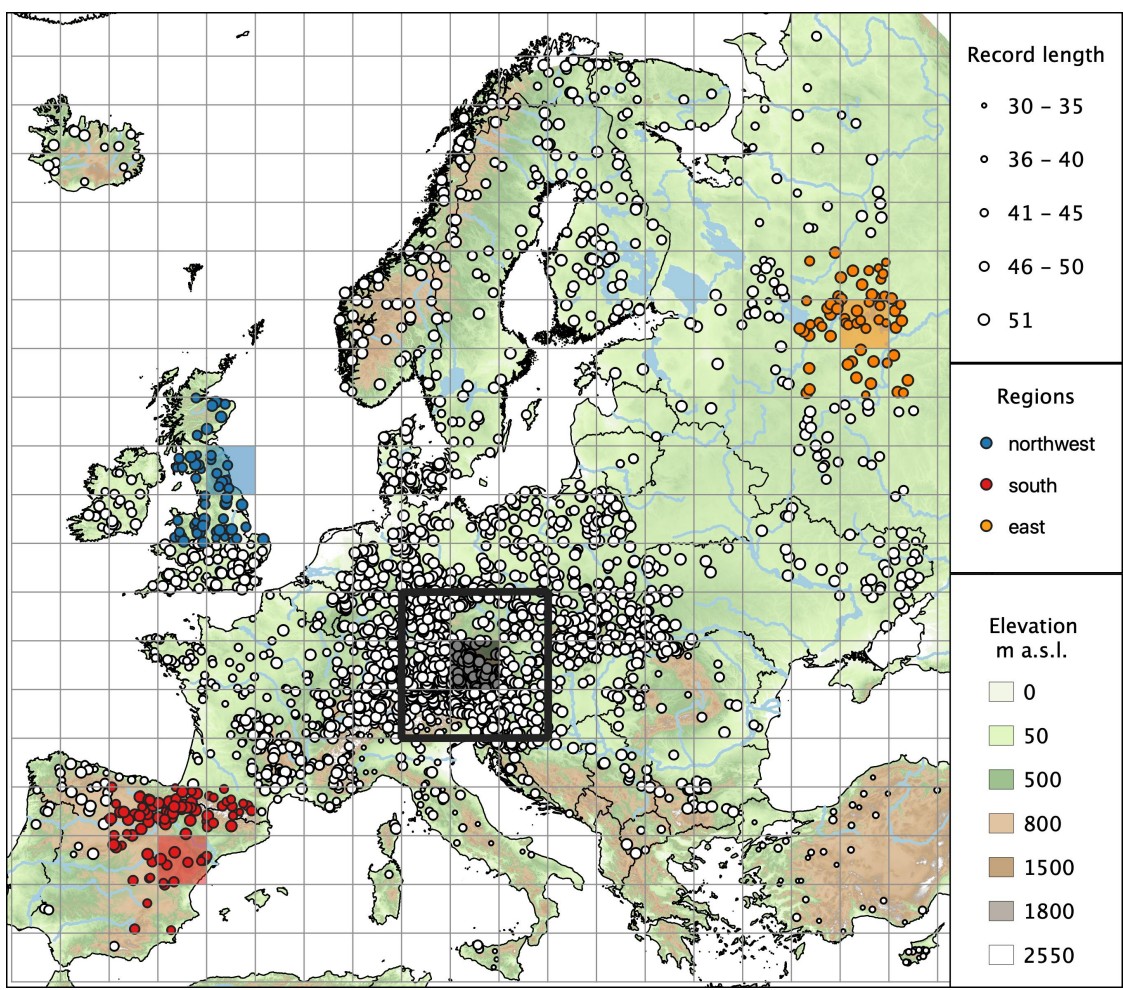

**Figure 1.** Location of 2370 hydrometric stations in Europe and regions considered in this study. The size of the circles is proportional to the length of flood records. The grid size is 200 km. The black bordered region shows the size of the spatial moving windows analysed in Sect. 3.2. It consists of nine cells, corresponding to 600 km × 600 km, whose central cell is black shaded. Three regions analysed in Sect. 3.3, respectively located in northwestern, southern and eastern Europe, are shown with coloured circles and the shaded regions represent their central cells.

## 2.4 Drivers of flood change

Because stations with substantial artificial alterations are not included in the database, in this study we consider three potential climatic drivers of flood change: (i) extreme precipitation, (ii) antecedent soil moisture and (iii) snowmelt. For each driver we obtain catchment-averaged time series, as described in detail in the following paragraphs, which are used as covariates in the regional model of Sect. 2.1. Unlike Viglione et al. (2016), scale dependence is here accounted for by the data, as we use local (i.e. catchment-averaged) covariates, and not directly into the model.





**Extreme precipitation**

Daily series of catchment-averaged precipitation between 1960 and 2010 are calculated for each hydrometric station from the daily gridded E-OBS precipitation and the catchment boundaries. For each station we identify a window around the average date of occurrence of floods $\bar{D}$, in which extreme precipitation is considered to be typically relevant for the generation of the annual peaks. The width of the window $w$ is set between 90 and 360 days and it is taken proportional to $1 - R$, with $R$ being the concentration of the date of occurrence around the average date, through the following equation:

$$w = 90 + (1 - R) \cdot 270 \, [days] \tag{10}$$

$\bar{D}$ and $R$ are obtained with circular statistics (see Appendix B). The window of dates is centred around $\bar{D}$, in a way that two thirds of the window occur before the average date of occurrence of floods (as shown in Fig. 2 for an example series in one example year). For each year in the period of interest, we calculate the 7-day maximum precipitation within the identified window (which varies between catchments but is fixed between years).

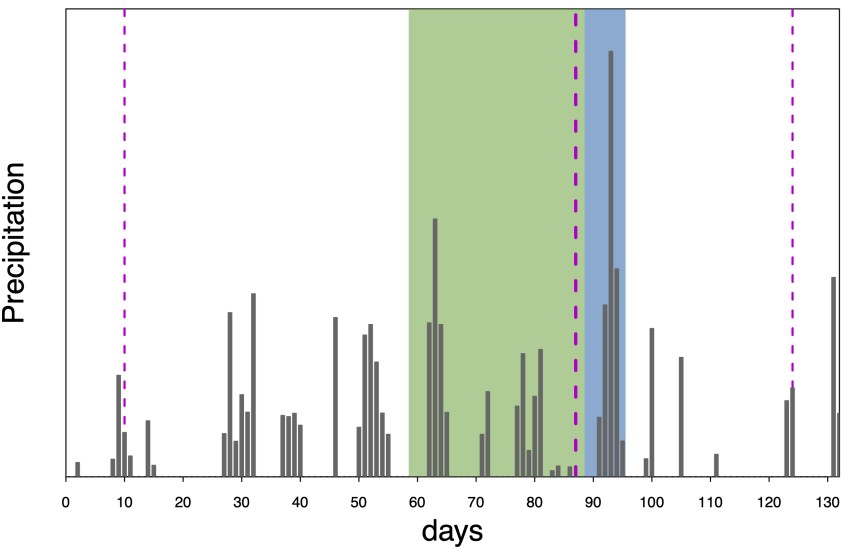

**Figure 2.** Procedure used to obtain the time series of extreme precipitation and antecedent soil moisture index. The figure shows the daily series of catchment-averaged precipitation for one example station in one example year. The thick dashed magenta line represents the average date of occurrence of annual floods for the example station and the two thin dashed lines indicate the window of dates around the average date of occurrence, where extreme (7-day maximum) precipitation is selected (blue area). The respective preceding 30-day precipitation (green area) is representative of the antecedent soil moisture. The procedure is repeated for every year in the period of interest and every hydrometric station.





**Antecedent soil moisture index**

An index of antecedent soil moisture is obtained from daily catchment-averaged precipitation. For each year and each station, we calculate the 30-day precipitation preceding the 7-day window identified for extreme precipitation above. We use this index (for brevity, hereinafter referred to as 'antecedent soil moisture') based on precipitation instead of modelled soil moisture, as in Blöschl et al. (2019), in order to more strongly rely on observational data.

**Snowmelt**

Similar to precipitation, daily series of catchment-averaged temperature between 1960 and 2010 are obtained for each hydrometric station. We calculate daily series of catchment-averaged snowmelt according to a simple degree-day model (Parajka and Blöschl, 2008) as a function of mean daily air temperature $T_A$ and precipitation $P$:

$$M = \begin{cases} 0 & for \ T_A < T_m \\ \min\left(\mathrm{DDF} \cdot (T_A - T_m); P_s\right) & for \ T_A \geq T_m \end{cases} \tag{11a}$$

$$P_S = \begin{cases} P & for \ T_A < T_s \\ P \cdot \frac{T_R - T_A}{T_R - T_S} & for \ T_S \leq T_A \leq T_R \\ 0 & for \ T_A > T_R \end{cases} \tag{11b}$$

Where $M$ and $P_s$ are the daily snowmelt depth and snow water equivalent storage, DDF is the degree day factor and $T_m$, $T_s$ and $T_R$ are the temperature thresholds that control the occurrence of melt, snow and rainfall, respectively. Here we assume $T_m = T_s = 0°$ C, $T_R = 2.5°$ C and $\mathrm{DDF} = 2.5$ mm day$^{-1}$ °C$^{-1}$ (Parajka and Blöschl, 2008; He et al., 2014). For each station, the time series of 7-day maximum snowmelt is obtained from daily snowmelt, using the same procedure illustrated above for the case of extreme precipitation.

As in Bertola et al. (2019), this study aims at attributing flood changes to the long-term evolution of the covariates rather than their year-to-year variability. For this reason, we smooth the annual series of the drivers with the locally weighted polynomial regression LOESS (Cleveland, 1979) using the R function *loess*. The subset of data over which the local polynomial regression is performed is 10 years (i.e. 10 data-points of the series) and the degree of the local polynomials is set equal to 0, which is equivalent to a weighted 10-year moving average.

## 2.5 A priori on model parameters

In the attribution analysis we use informative priors on the parameters controlling the relationship between flood and covariate changes (see Bertola et al., 2019). This is done because we do not want to use the time patterns of the covariates $X_i$ only to discriminate between drivers, which may lead to spurious correlations, but to hydrologically 'inform' the attribution analysis. Therefore, we set a priori constraints on the model parameters, based on qualitative reasoning and on prior literature. Given the





covariates considered in this study, the elasticities of flood quantiles to the drivers (defined in Eq. 5) are expected to be positive (i.e. we expect the changes in $X_i$ and $q_T$ to have the same sign). For T=2 and 100 years, this translates respectively into:

$$\alpha_{2_i} > 0 \tag{12a}$$

$$\alpha_{2_i} + \alpha_{g_i}\left(1 - \frac{1}{1 + x'_{100}}\right) > 0 \tag{12b}$$

Eq. 12a represents the lower limit for the elasticity parameters of $q_2$. The lower limit for $\alpha_{g_i}$ is obtained from Eq. 12b and depends on $\alpha_{2_i}$ and on the growth factor:

$$\alpha_{g_i} > -\frac{\alpha_{2_i}}{1 - \frac{q_2}{q_{100}}} \tag{13}$$

For simplicity, we assume $q_{100} = 2q_2$ as a reasonable approximation valid for Europe (Blöschl et al., 2013; Alfieri et al., 2015), and we simplify Eq. 13 to:

$$\alpha_{g_i} > -2\alpha_{2_i} \tag{14}$$

The prior distributions of $\alpha_{2_i}$ and on $\alpha_{g_i}$ are modelled as normal distributions $\mathcal{N}(0,2)$ with truncated lower tail, as summarised in Tab. 1. For the remaining parameters we set an improper uniform prior distribution.

| Parameter | Meaning | Lower limit | Distribution type |
|:---:|:---:|:---:|:---:|
| $\alpha_{2_1}$ | Elasticity of $q_2$ to $X_1$ | 0 | Truncated normal |
| $\alpha_{2_2}$ | Elasticity of $q_2$ to $X_2$ | 0 | Truncated normal |
| $\alpha_{2_3}$ | Elasticity of $q_2$ to $X_3$ | 0 | Truncated normal |
| $\alpha_{g_1}$ | Elasticity of $x'_{100}$ to $X_1$ | $-2\alpha_{2_1}$ | Truncated normal |
| $\alpha_{g_2}$ | Elasticity of $x'_{100}$ to $X_2$ | $-2\alpha_{2_2}$ | Truncated normal |
| $\alpha_{g_3}$ | Elasticity of $x'_{100}$ to $X_3$ | $-2\alpha_{2_3}$ | Truncated normal |

**Table 1.** A priori on model elasticity parameters

## 2.6 Regional analyses

Following the spatial moving window approach of Bertola et al. (2020), we identify several regions of size 600 km × 600 km across Europe, which overlap by 200 km in both directions. We fit the regional flood change model of Sect. 2.1 to pooled flood and covariate data of sites within each region. The resulting 200 km x 200 km grid cells are shown in Fig. 1 and each of the considered regions is composed of nine adjacent cells, (e.g. the black bordered region in Fig. 1). In each region, we estimate the elasticity of $q_2$ and $q_{100}$ to the drivers $X_i$ and the contribution of each driver to flood changes, obtained by multiplying the elasticity by the average driver trend in the region (Eq. 6). In regions where the average 7-day maximum snowmelt is less than 2 mm/day, only extreme precipitation and antecedent soil moisture are considered as potential drivers (i.e. Eq. 4a and 4b are modified by removing the contribution of $X_3$). The resulting elasticity and contribution are plotted in the central 200 km x 200





km cell of the region (e.g. the shaded cell in the black bordered region in Fig. 1). The rationale of the homogeneity assumption is that the spatial windows, given their size, have rather homogeneous climatic conditions (and hence flood generation processes

and processes driving flood changes) relative to the overall variability within Europe. The results of this analysis are shown in Sect. 3.2. In Sect. 3.3, the elasticities of flood quantiles to the drivers and their contributions to flood change are further analysed as a function of the return period, for three regions located respectively in northwestern, southern and eastern Europe (see Fig. 1).

## 3 Results

### 3.1 Drivers of flood change

Time series of catchment-averaged (i) extreme precipitation, (ii) antecedent soil moisture and (iii) snowmelt are obtained for each hydrometric station for the period 1960-2010, as described in Sect. 2.4. Figure 3 shows maps of the mean value and the change of these drivers for each station in the period of interest. Extreme precipitation (Fig. 3a) exhibits its largest mean values in central and western Europe, particularly in the Alpine region and on the western Atlantic coast. Positive changes of extreme

precipitation are observed in the Alpine region, northwestern and central Europe, Scandinavia and Poland; negative changes are observed in southern countries and in few spots in central Europe (Fig. 3d). Similar spatial patterns appear for antecedent soil moisture (Fig. 3b and 3e), but the negative changes tend to be more widespread and with stronger (negative) magnitude. Mean snowmelt is largest in northeastern Europe and in the Alpine region (Fig. 3c). Its changes are mostly negative across all Europe, with the exception of the very North and few isolated spots (Fig. 3f).





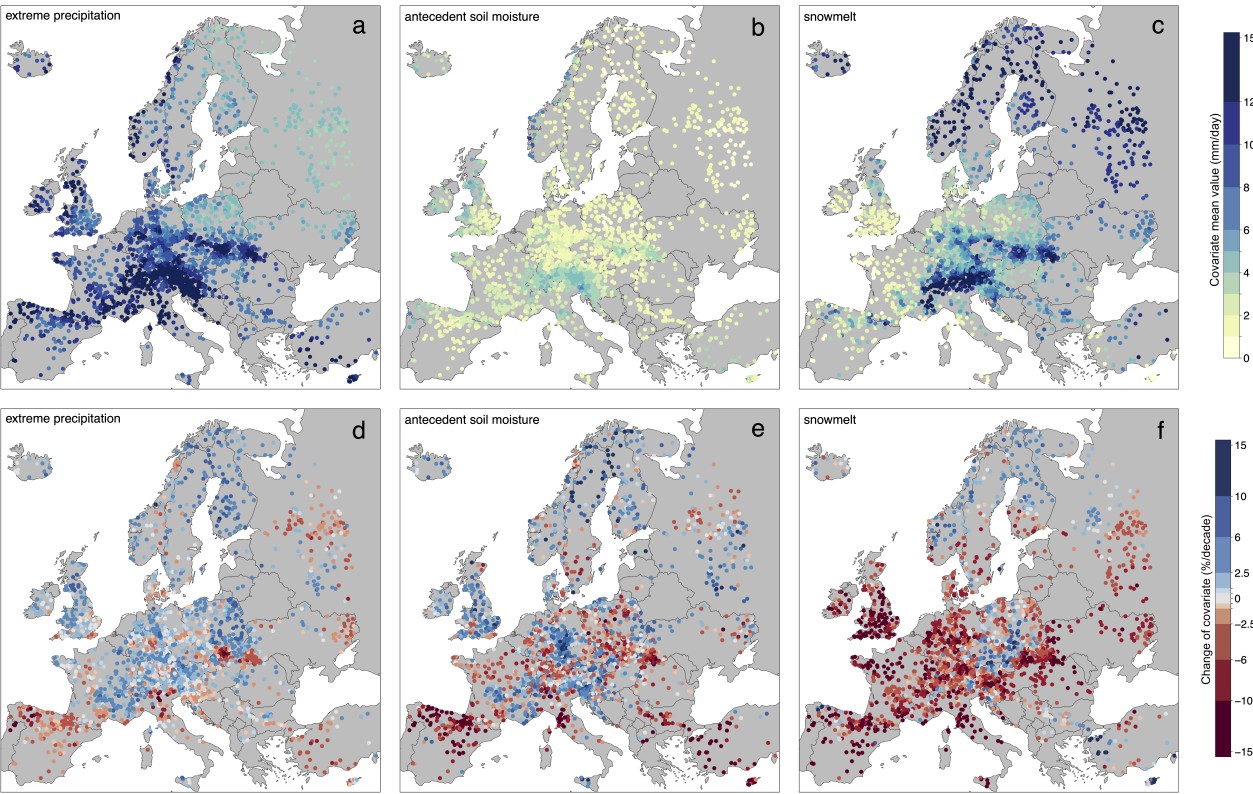

**Figure 3.** Mean value and change of catchment-averaged extreme precipitation (a,d), antecedent soil moisture (b,e) and snowmelt (c,f) for each station over the period 1960-2010.

## 3.2 Contributions of the drivers to flood change across Europe

The obtained time series of catchment-averaged extreme precipitation, antecedent soil moisture and snowmelt are used as covariates in the regional driver-informed model. Figure 4 shows maps of the elasticity of the 2-year flood $q_2$ and the 100-year flood $q_{100}$ to each of the three drivers, as defined in Eq. 5, resulting from fitting the regional model to the pooled flood and covariate data in moving windows across Europe. The value of the posterior median of the elasticities is shown together with the 90% credible bounds. The elasticity of $q_2$ to extreme precipitation (Fig. 4a) is large (0.6 to 1.5) in western, central and southern Europe and lower values (0 to 0.25) are observed in northeastern Europe. Similar values of elasticity to extreme precipitation are observed for the 100-year flood across Europe (Fig 4b), with small differences in northeastern Europe. This means that the elasticity of flood quantiles to extreme precipitation does not vary much with return period. In contrast, the elasticity of flood quantiles to soil moisture decreases with return period (Fig. 4b and 4e) and it is largest in southern Europe (0.25 to 0.6). Overall, the elasticities of $q_2$ and $q_{100}$ to soil moisture are smaller than those to extreme precipitation. The elasticity of floods to snowmelt is largest in northeastern Europe (Fig. 4c and 4d), where values above 1 are observed (i.e. a change of 1% in





snowmelt translates into a change in flood quantiles larger than 1%). In northeastern Europe the elasticities of $q_2$ and $q_{100}$ to snowmelt are similar, while in central Europe and the Balkans they decrease with the return period.

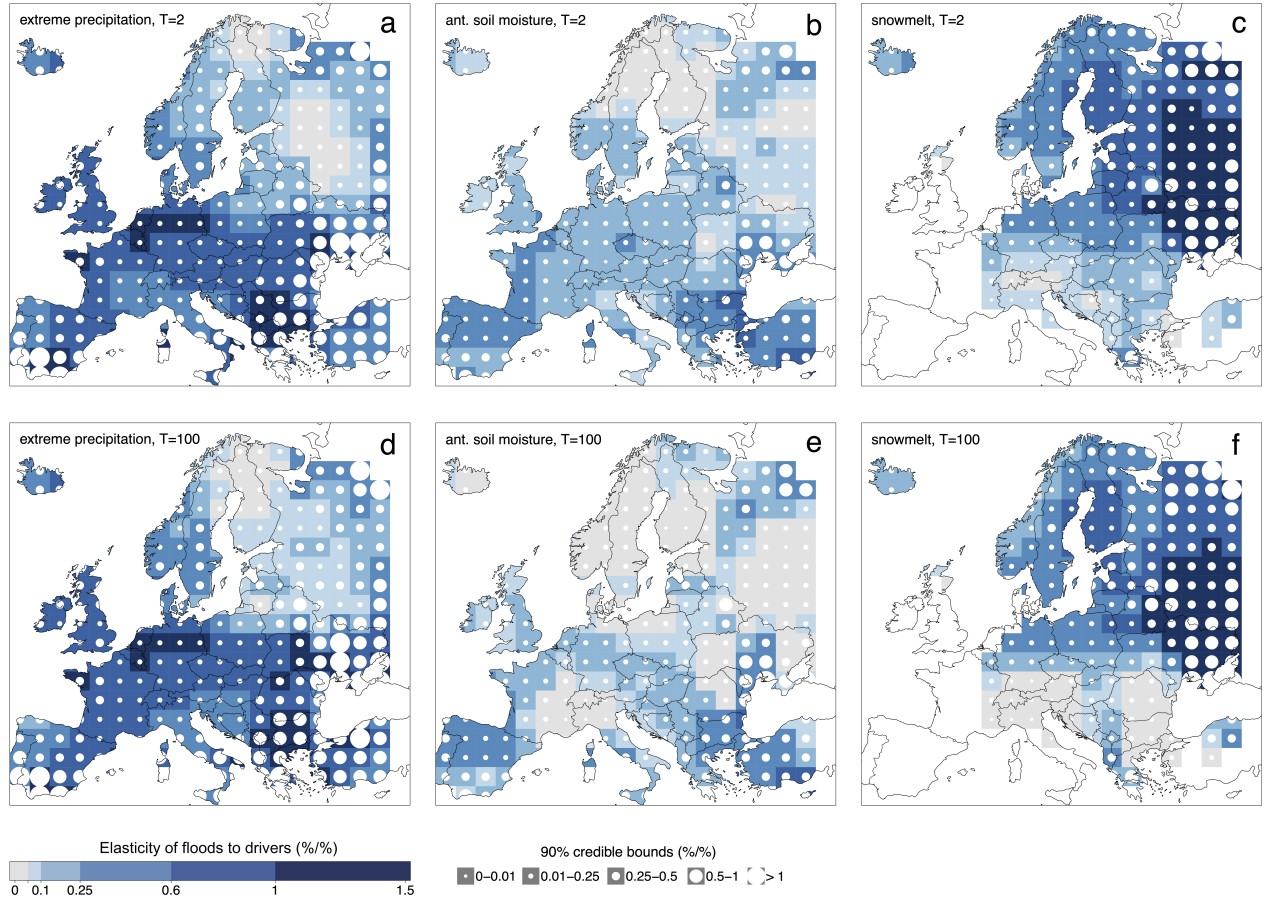

**Figure 4.** Elasticity of the 2-year flood $q_2$ (upper panels) and the 100-year flood $q_{100}$ (lower panels) to extreme precipitation (a, d), antecedent soil moisture (b, e) and snow melt (c, f). The median value of the posterior distribution of the elasticity is shown in each region with colours and the size of the white circles is proportional to the respective 90% credible bounds. The maps are shown for hypothetical catchment area of 1000 km$^2$

Figure 5 shows maps of the contributions of each of the three drivers to changes in $q_2$ and $q_{100}$, as defined in Eq. 6. They are obtained by multiplying the elasticities of flood quantiles to the drivers by the average changes (in % per decade) in the drivers in each region over the period 1960-2010 (Eq. 6). They represent the change in flood quantiles, in % per decade, caused by the change in a specific driver. Extreme precipitation (Fig. 5a and 5d) contributes positively to flood changes in northwestern and central Europe, and negatively in southern and eastern Europe. The absolute value of the contributions of extreme precipitation appears to slightly decrease when moving from $q_2$ to $q_{100}$. Antecedent soil moisture contributes mostly to negative flood changes in southern Europe (Fig. 5b and 5e) and the magnitude of this contribution decreases with the return





period. The contributions of snowmelt to changes in $q_2$ and $q_{100}$ are predominantly negative and marked in Eastern Europe, with small differences towards smaller contributions in absolute values with return period (Fig. 5c and 5f). In contrast, snowmelt contributes to positive flood changes in Scandinavia, and to a lesser extent for $q_{100}$ than for $q_2$. Overall the uncertainties associated with the contribution of the drivers to changes in $q_{100}$ do not seem to increase much compared to $q_2$.

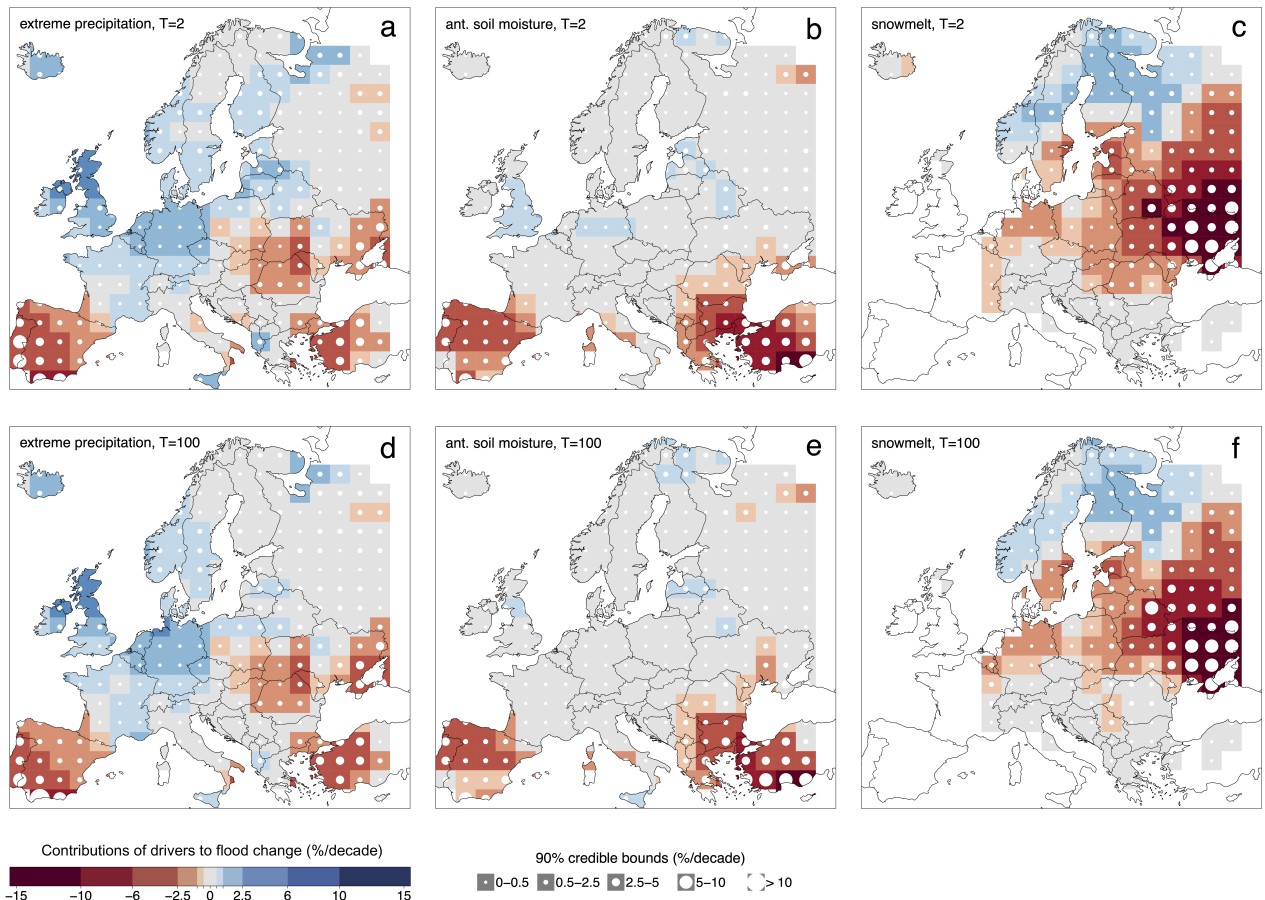

**Figure 5.** Same as Fig. 4, but for contributions of extreme precipitation (a, d), antecedent soil moisture (b, e) and snow melt (c, f) to changes in $q_2$ and $q_{100}$.

In order to further investigate the differences in terms of (absolute) contributions of the drivers to changes in large (i.e. $q_{100}$) versus small floods (i.e. $q_2$), we compute for each driver the ratio between these two quantities (Fig. 6). In the case of extreme precipitation (Fig. 6a), the ratio between its contributions to changes in $q_{100}$ and $q_2$ is between 0 and 1 in the Atlantic region, Spain, Italy, the Balkans, southern Germany, Austria and Finland, i.e., in these regions the contribution of extreme precipitation to changes in $q_{100}$ is smaller, in absolute value, compared to changes in $q_2$. In southern France, eastern Europe and Turkey the opposite is observed (i.e. the ratio is larger than 1). Antecedent soil moisture and snowmelt generally contribute less to changes





in $q_{100}$ compared to $q_2$ (Fig. 6b and 6c). Large uncertainties in the ratio of elasticities are observed in northeastern Europe, in the case of extreme precipitation and antecedent soil moisture (Fig. 6a and 6b), and in southern Europe, in the case of snowmelt (Fig. 6c), and they result from values of the contribution of the drivers to $q_2$ that are close to zero in these regions (see Fig. 5).

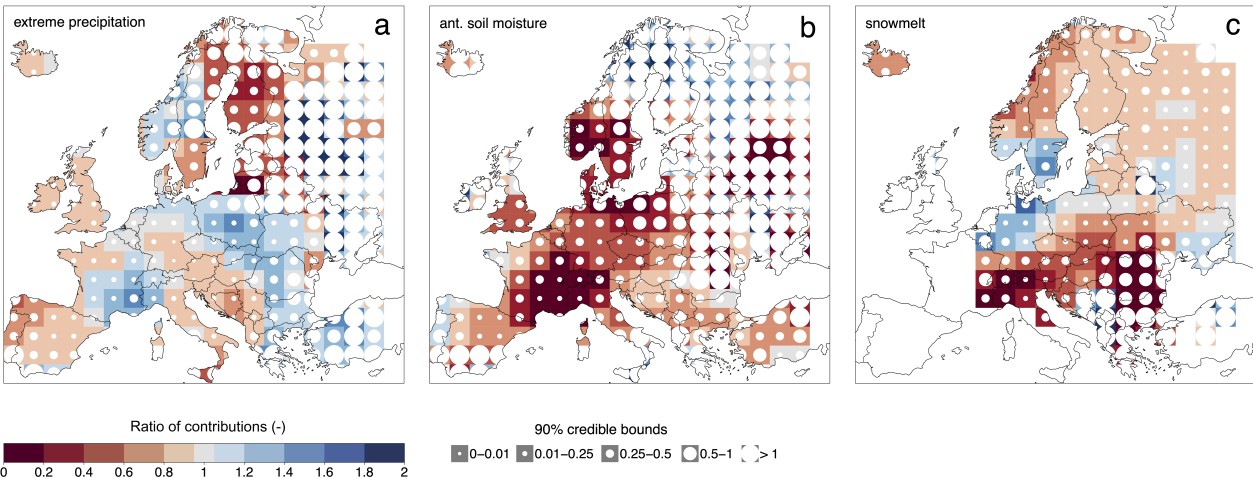

**Figure 6.** Same as Fig. 4, but for the ratios of the contributions of extreme precipitation (a), antecedent soil moisture (b) and snow melt (c) to changes in $q_{100}$ relative to $q_2$. Values below 1 (red colour) indicate that the contribution of the driver to $q_{100}$ is smaller than the contribution to $q_2$; values above 1 (blue colour) indicate that the contribution of the driver to $q_{100}$ is larger than the contribution to $q_2$.

Finally, for each region we obtain the relative contribution of the three drivers to changes in $q_2$ and $q_{100}$, as defined in Eq. 8
(Fig. 7). The relative contribution of extreme precipitation is the largest of all the drivers in most of western and central Europe for both $q_2$ and $q_{100}$ (Fig. 7a and 7d). The relative contribution decreases somewhat with return period in northwestern Europe, while the opposite is the case in the South. In southern Europe antecedent soil moisture has the largest relative contribution to changes in $q_2$ (Fig. 7b) and its relative importance tends to decrease for more extreme floods (Fig. 7e). The relative contribution of snowmelt to flood changes clearly prevails in eastern Europe, with slightly decreasing strength for the higher return period.





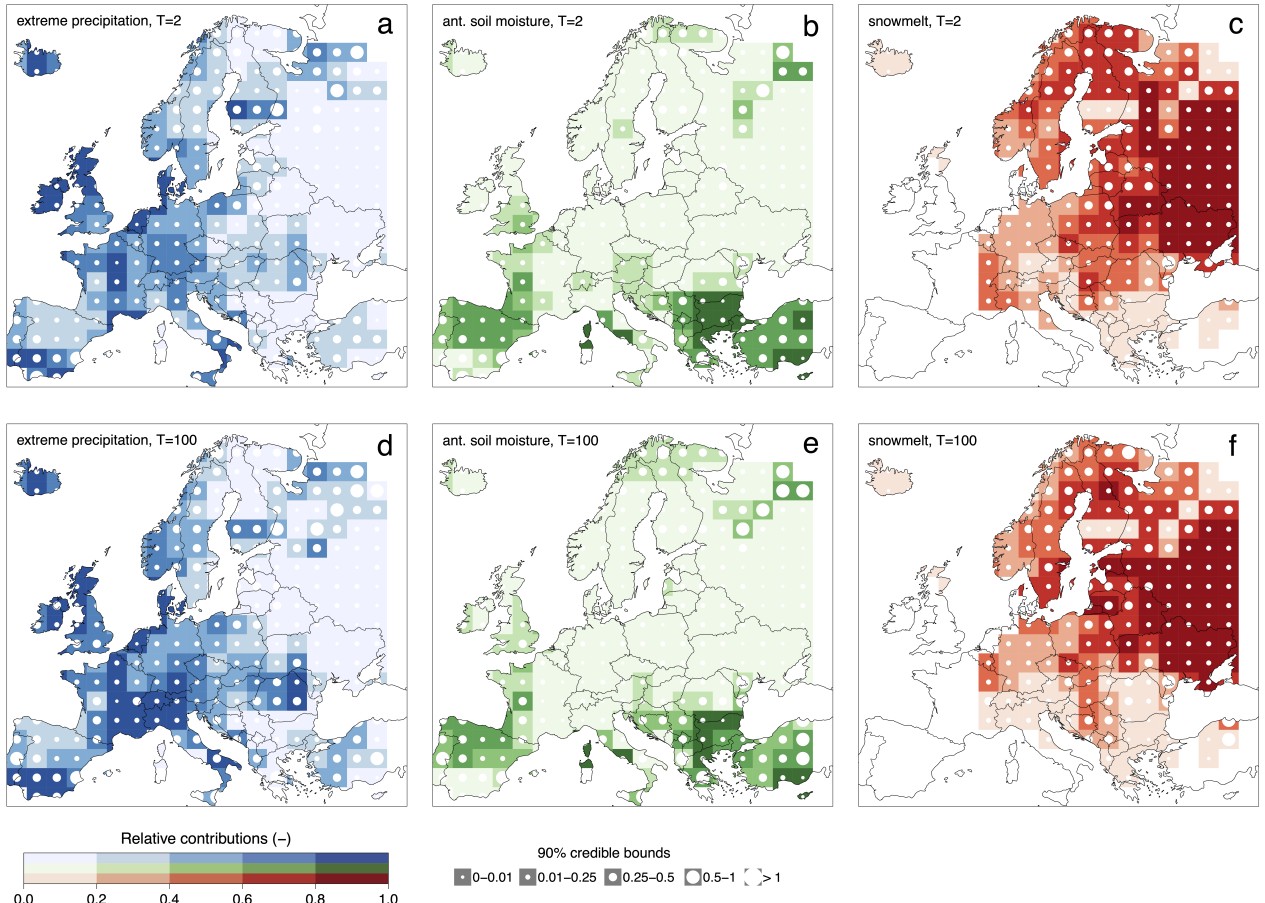

**Figure 7.** Same as Fig. 4, but for relative contributions of extreme precipitation (a, d), antecedent soil moisture (b, e) and snow melt (c, f) to changes in $q_2$ and $q_{100}$.

## 3.3 Contributions to flood change of the drivers in northwestern, southern and eastern Europe

In this section we select three example regions among those analysed in Sect. 3.2, located respectively in northwestern, southern and eastern Europe (see Fig. 1). For these three regions we further show in Fig. 8 the elasticities of floods to the drivers (first row), the contributions (second row) and relative contributions (third row) of the drivers to flood change, as a function of the return period. In the regions located in northwestern and southern Europe, snowmelt is excluded from the potential drivers as it does not represent a relevant process for most of the catchments in these regions (see Fig. 3c). In the region in northwestern Europe extreme precipitation and antecedent soil moisture contribute positively to flood change, with extreme precipitation representing the most important driver. Its contribution to flood trends decreases with return period, while the contribution stays almost constant in the case of antecedent soil moisture (Fig. 8d and 8g). In the region in southern Europe extreme precipitation and antecedent soil moisture represent both important drivers. The elasticity of floods to extreme precipitation





is larger than that to antecedent soil moisture (Fig. 8b). However, antecedent soil moisture contributes (negatively) to a larger extent to flood changes for small return periods (i.e. T=2-10 years). Its contribution decreases in absolute values with increasing return period (Fig. 8e). For more extreme events (T>10 years) the relative contribution of extreme precipitation increases and becomes comparable to that of antecedent soil moisture (Fig. 8h). In the region in eastern Europe snowmelt is clearly the dominant driver at all return periods (Fig. 8c, 8f and 8i).





**Figure 8.** Contributions of drivers to flood changes as a function of the return period in three regions (columns), respectively located in northwestern, southern and eastern Europe. Elasticity of floods to the drivers (a, b, c), contribution (d, e, f) and relative contribution (g, h, i) of the drivers to flood change are shown in the rows. The thick lines and the shaded areas represent respectively the median and the 90% credible intervals of their posterior distributions. The results are shown for hypothetical catchment area of 1000 km$^2$.



## 4 Discussion and conclusions

In this study, we attribute the changes in flood discharges that have occurred in Europe during the period 1960-2010 (Blöschl et al., 2019; Bertola et al., 2020) to potential drivers as a function of the return period, while previous detection and attribution studies have generally focused on the mean flood behaviour. In particular, we compare the relative contribution of extreme precipitation, antecedent soil moisture and snow melt to changes in the median and the 100-year flood. The attribution study is framed in terms of a non-stationary flood frequency analysis and the parameters of the distribution are estimated in a regional context with Bayesian inference.

### 4.1 Is it possible to identify the relative contributions of different drivers to $q_{100}$ changes as compared to $q_2$ changes?

Our results suggest that in northwestern and eastern Europe, changes in small and large floods are driven mainly by one single driver, which dominates at all return periods. In northwestern Europe, extreme precipitation contributes to changes in both $q_2$ and $q_{100}$ for the most part and the contribution of antecedent soil moisture is of secondary importance. Similarly, in eastern Europe, snowmelt clearly drives flood changes at all return periods. In southern Europe both antecedent soil moisture and extreme precipitation significantly contribute to flood changes and their relative importance depends on the return period. Antecedent soil moisture contributes the most to changes in small floods (i.e. T=2-10 years), while the two drivers contribute with comparable magnitude to changes in more extreme events (T>10 years). Given the relative driver contributions and their credible bounds obtained in the analysis, the findings suggest that is indeed possible to identify the relative contributions to $q_2$ and $q_{100}$ clearly.

### 4.2 What is the nature (sign and magnitude) of these contributions?

The contribution of extreme precipitation is positive in northwestern Europe (about 3.3 to 2.8% per decade in Fig. 8) and decreases slightly with return period in the region analysed in Sect. 3.3. In contrast, in the region selected in southern Europe extreme precipitation contributes to 37 to 45% of the negative flood changes (corresponding to -2.2 to -1.8% per decade), depending on the return period. The contribution of antecedent soil moisture is negative in southern Europe and decreases in absolute value (from -3.8 to -2.3% per decade) with the return period in the analysed region. Finally, in eastern Europe snowmelt strongly contributes to negative flood changes in a similar way at all return periods (about -3% per decade for the region in Sect. 3.3). The sum of the contributions of the drivers of Fig. 5 is in overall agreement with the flood change patterns and trend magnitudes found by Blöschl et al. (2019) and Bertola et al. (2020), with the exception of Scandinavia, where the contributions of the drivers are all positive or close to zero, while mostly moderate negative flood trends were observed in previous studies (Blöschl et al., 2019; Bertola et al., 2020). This discrepancy points to other drivers not accounted for in the presented model, such as river regulation effects (Arheimer and Lindström, 2019), or non linear relationships between the drivers not captured by the model.



## 4.3 Model assumptions

Prior information on the elasticities is used in order to 'inform' the attribution analysis, based on hydrological reasoning and the literature. Specifically, the prior distribution of the elasticities of $q_2$ and $q_{100}$ to the drivers are assumed positive. This is because any changes in the considered covariates are expected to translate into flood changes with the same sign. In practice, the prior distribution of the elasticity of $q_{100}$ is reflected in a lower bounded prior distribution on the elasticity of the growth factor $x'_{100}$, which depends on the ratio between $q_{100}$ and $q_2$ (Sect. 2.5). For simplicity, we assume this ratio approximately equal to 2. This assumption is reasonably valid for humid catchments (see e.g., Blöschl et al., 2013) and is in overall agreement with flood maps of the mean annual flood and $q_{100}$ in Europe presented by Alfieri et al. (2015). However, in arid regions, larger values of this ratio (e.g. 4, see Blöschl et al., 2013) would be more appropriate (corresponding to stricter priors on the elasticity of the growth factor) because the flood frequency curves tend to be steeper.

We fitted the change model of Sect. 2.1 to the pooled flood and covariate data of several regions across Europe, where elasticities of flood quantiles to their drivers are assumed homogeneous. This assumption is reasonable because of the spatial proximity of the catchments that is reflected into similar climatic conditions, flood generation processes and processes driving flood changes. The attribution analysis is thereby performed at the regional scale, where average regional contributions of the decadal changes in the drivers to average regional trends in flood quantiles are estimated. Figure 8 shows the contributions of the drivers to flood changes as a function of the return period for three regions selected respectively in northwestern, southern and eastern Europe. Similar results would be obtained by fitting the model to larger regions over northwestern, southern and eastern Europe, that present comparatively homogeneous flood regime changes and processes driving flood changes (e.g. the three macro-regions in Blöschl et al., 2019; Bertola et al., 2020).

Overall the obtained uncertainties associated with the contribution of the drivers to changes in $q_{100}$ do not seem to increase much compared to $q_2$, while a relevant increase would be reasonably expected. These results are valid under the assumption of the adopted model (i.e. Gumbel distribution) which may be too stringent. The model assumptions could be relaxed (e.g. adopting a Generalized Extreme Value distribution) in order to allow for larger model flexibility.

Spatial cross-correlation of floods at different sites is taken into account through an approach based on a magnitude adjustment to the likelihood. This results in larger uncertainties of the posterior distribution of the estimated parameters, compared to the case where floods are considered spatially independent.

As already noted, one of the main assumptions in our analysis is that the three drivers, i.e., extreme precipitation, soil moisture and snowmelt, are the only candidates for explaining river flood changes. The effects of other drivers not accounted for in this study, such as land cover change or river regulation, are probably not very large at the scale of Europe as we are focusing on catchments with minimum alteration. However, in contexts where anthropogenic alterations are important it will be useful to extend the analysis for such effects. This attribution analysis may be repeated with catchment (e.g. land-use or land-cover changes) and river drivers (e.g. construction of reservoirs in the catchment) in addition to atmospheric covariates, if detailed information about changes in land-use/land-cover and river structures were available for European catchments and flood data of affected stations were collected.




## 4.4 Conclusions

This study complements recent research on past changes in European floods by formally attributing the detected trends to potential drivers (i.e., extreme precipitation, antecedent soil moisture and snowmelt) as function of return period. The proposed method allows to identify the relative contribution of different drivers to changes in flood quantiles and to estimate the sign and magnitude of these contributions. The results show that in northwestern and eastern Europe changes in both the 2-year and the 100-year flood are driven by a single driver only (i.e. respectively extreme precipitation and snowmelt), while in

southern Europe two drivers contribute to flood changes (i.e. soil moisture and extreme precipitation), with different relative contributions depending on the return period. Even though this study focuses on observed flood changes, the understanding of past processes is a fundamental step for the prediction of flood changes in future climate scenarios.

*Data availability.* The flood discharge data used in this paper can be obtained from the Supplement of Blöschl et al. (2019) and are accessible at https://github.com/tuwhydro/europe_flood. Data regarding catchment areas belong to different institutions listed in Extended Data Table 1,

Blöschl et al. (2019). Catchment boundaries from CCM River and Catchment database is available at https://ccm.jrc.ec.europa.eu/php/index.php?action=view&id=23. E-OBS gridded precipitation and temperature dataset is available at https://www.ecad.eu/download/ensembles/download.php.

## Appendix A: Adjustment to the likelihood

Under the assumption of spatial independence of the data, the asymptotic distribution of the maximum likelihood estimator $\hat{\theta}$

of the independence likelihood is: $\hat{\theta} \sim N\left(\theta^0, n^{-1}H^{-1}VH^{-1}\right)$, where $\theta^0$ is the true value of $\theta$ and $H^{-1}VH^{-1}$ is the modified covariance matrix, where $H = -\mathbb{E}\nabla^2 l\left(\theta^0, \mathbf{y}\right)$ and $V = Cov\nabla l\left(\theta^0, \mathbf{y}\right)$. If the assumption of spatial independence is correct, we have that $H = V$. In Sect 2.2 we described an approach, proposed by Ribatet et al. (2012), that enables to account for spatial cross-correlation in spatial datasets and consists in an overall adjustment to the likelihood. In this analysis we adopted a magnitude adjustment, through a factor $k$ (Eq. 9). Ribatet et al. (2012) proposed to estimate $k$ by setting:

$$k = \frac{p}{\sum_{i=1}^{p}\lambda_i} \tag{A1}$$

where $p$ is the number of parameters in the independence likelihood and $\lambda_i$ are the eigenvalues of the matrix $H^{-1}V$. The matrix $H$ is approximated by the observed information matrix $\nabla^2 l\left(\hat{\theta}, \mathbf{y}\right)$ and $V$ is estimated by decomposing the likelihood into independent yearly contributions.





## Appendix B: Seasonality of floods

As in Blöschl et al. (2017), the average date of occurrence of floods $\bar{D}$ and the concentration $R$ of the date of occurrence around the average date are obtained with circular statistics, by conversion of the date of occurrence of a flood in the year $i$ into an angular value $D_i$:

$$\bar{D} = \begin{cases} \tan^{-}1\left(\frac{\bar{y}}{\bar{x}}\right) \cdot \frac{\bar{m}}{2\pi} & \bar{x} > 0, \bar{y} \geq 0 \\ \tan^{-}1\left(\frac{\bar{y}}{\bar{x}} + \pi\right) \cdot \frac{\bar{m}}{2\pi} & \bar{x} \leq 0 \\ \tan^{-}1\left(\frac{\bar{y}}{\bar{x}} + 2\pi\right) \cdot \frac{\bar{m}}{2\pi} & \bar{x} > 0, \bar{y} \leq 0 \end{cases} \tag{B1a}$$

$$R = \sqrt{\bar{x}^2 + \bar{y}^2} \tag{B1b}$$

with:

$$\bar{x} = \frac{1}{n}\sum_{i=1}^{n}\cos\theta_i \tag{B2a}$$

$$\bar{y} = \frac{1}{n}\sum_{i=1}^{n}\sin\theta_i \tag{B2b}$$

$$\theta_i = D_i \cdot \frac{2\pi}{m_i} \tag{B2c}$$

Where $n$ is the number of peaks registered at that station, $m_i$ is the number of days in the year $i$ and $\bar{m}$ is the average number

of days per year. When floods occur equally throughout the year $R = 0$, while $R = 1$ when floods always occur on the same date.

*Author contributions.* All co-authors designed the overall study. MB performed the analysis and prepared the paper. All co-authors contributed to the interpretation of the results and writing of the paper.

*Competing interests.* The authors declare that they have no conflict of interest

*Acknowledgements.* The authors would like to acknowledge funding from the European Union's Horizon 2020 Research and Innovation Programme under the Marie Skłodowska-Curie grant agreement no. 676027, the FWF Vienna Doctoral Programme on Water Resource Systems (W1219-N28), the Austrian Science Funds (FWF) "SPATE" project I 3174 and the German Research Foundation (DFG; grant no. FOR 2416).

We acknowledge the E-OBS dataset from the EU-FP6 project UERRA (http://www.uerra.eu) and the Copernicus Climate Change Service,

and the data providers in the ECA&D project (https://www.ecad.eu).



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
