# Peer review of "Do small and large floods have the same drivers of change? A regional attribution analysis in Europe"

_Hydrology and Earth System Sciences, 2020_

## Short Comment (SC1) · 19 Aug 2020

The work is interesting and important as well. Nevertheless, I doubt that you can come up with some conclusions for entire Europe, especially for the south of Europe. Most of the data come from central Europe, which more less has the same climate conditions. There is a lack of representative data from the most diverse climate and vulnerable regions, namely south of Europe. Surprisingly, although there are data available, even long records, more than half of the Iberian Peninsula is left out, is not considered in the analysis. Also, western Balkans almost out. Those regions are expected to be profoundly impacted by climate changes, why you did not consider

hydrological data from those regions? I think it is not a proper way to extrapolate and/or interpolate results in such disproportionate data consideration. I wonder, is this a perfect work? Frankly speaking, I would say not, and it is ok because science is evolving and improving by time. Nevertheless, the authors do not talk about the limitations of the work. I believe this is a critical issue to be discussed in the manuscript. I see you have a section called '' Model assumptions", I think this is a bit confusing because not every assumption can be a limitation. I would suggest to move it in the method section, not in the discussion. The discussion section needs to be elaborated more; the current version does not cover all aspects of the study. Conclusions also need some further elaboration.

---

## Short Comment (SC2) · 26 Aug 2020

Thanks for the clarification! I understand the limitations.

---

## Author Comment (AC1) · 26 Aug 2020

We would like to thank Alban Kuriqi for his comment.

Alban Kuriqi is suggesting that there is a lack of representative data from the south of Europe. In this study we have analyzed the flood database presented in Blöschl et al. (2019). It consists of 2370 flood records from 33 European countries. In southern Europe, as defined in Bertola et al. (2020), we have a total of 458 stations. Stations affected by strong artificial alterations (such as large reservoirs in the proximity of the gauges) are not included in this database. Because of the large number of reservoirs in the south of Europe, particularly on the Iberian Peninsula, the density of stations

is lower than in central Europe. The uneven distribution of stations in Europe will be acknowledged more explicitly in the revised paper.

The uneven distribution of stations across Europe is accounted for in the analysis by the credible bounds (represented by white circles) shown in Figures 4, 5 and 7. In data-scarce regions the credible bounds tend to be larger, i.e. the attribution results are shown with larger uncertainties. We will further emphasize this aspect in the manuscript.

Alban Kuriqi is also suggesting to move section 4.3 that discusses the influence of the model assumptions on the results to the methods section. While the model assumptions themselves are described in section 2, we consider the current position of section 4.3 appropriate. In order to enhance its clarity, we will change the heading of section 4.3 "Model assumption" into "Discussion of model assumptions".

References:

Blöschl, G. et al. (2019) 'Changing climate both increases and decreases European river floods', Nature, 573(7772), pp. 108–111. doi: 10.1038/s41586-019-1495-6.

Bertola, M., Viglione, A., Lun, D., Hall, J., & Blöschl, G. (2020). Flood trends in Europe: are changes in small and big floods different?. Hydrology & Earth System Sciences, 24(4).

―――――――――――――――――――

---

## Short Comment (SC3) · 7 Sep 2020

This is a very interesting study documenting the drivers of floods in Europe.

I have a small comment on line 195, about the use of 30-day antecedent rainfall. How are the results sensitive to the choice of this 30-day average ? With basin sizes ranging from 5 to 100 000 km2, it is likely that 30-day antecedent rainfall is a rough approximation of the actual soil moisture conditions for this range of basin sizes. For very small basins, 30-day rainfall could be enough, but maybe larger accumulations periods could be more adapted for larger basins. Why not consider an API, as in Woldemeskel and Sharma (2016) ? The question behind my question is: do we under-estimate the

effects of antecedent soil moisture by using only a 30-day antecedent rainfall ?

Woldemeskel, F., and Sharma, A. (2016), Should flood regimes change in a warming climate? The role of antecedent moisture conditions, Geophys. Res. Lett., 43, 7556–7563, doi:10.1002/2016GL069448.

Note: line 65, Tramblay et al. 2013 is wrong, the correct reference is Tramblay et al., 2019 (https://doi.org/10.5194/hess-23-4419-2019).

Yves Tramblay

---

## Author Comment (AC2) · 16 Sep 2020

**Short comment SC3:**

This is a very interesting study documenting the drivers of floods in Europe. I have a small comment on line 195, about the use of 30-day antecedent rainfall. How are the results sensitive to the choice of this 30-day average ? With basin sizes ranging from 5 to 100 000 km2, it is likely that 30-day antecedent rainfall is a rough approximation of the actual soil moisture conditions for this range of basin sizes. For very small basins, 30-day rainfall could be enough, but maybe larger accumulations periods could be more adapted for larger basins. Why not consider an API, as in Woldemeskel and Sharma (2016) ? The question behind my question is: do we under-estimate the effects of antecedent soil moisture by using only a 30-day antecedent rainfall ?

Woldemeskel, F., and Sharma, A. (2016), Should flood regimes change in a warming climate? The role of antecedent moisture conditions, Geophys. Res. Lett., 43, 7556–7563, doi:10.1002/2016GL069448.

Note: line 65, Tramblay et al. 2013 is wrong, the correct reference is Tramblay et al., 2019 (https://doi.org/10.5194/hess-23-4419-2019).

**Yves Tramblay**

**Reply to Short comment SC3:**

We would like to thank Yves Tramblay for his comment. In this study we used 30-day antecedent precipitation (AP30) as an indicator of antecedent moisture condition in the catchments because temporal windows of 30 days or less are typically used in the literature. In fact, antecedent precipitation (AP) refers to a wide range of temporal windows, from one hour to 30 days, and no explicit guidelines on this duration are available (Ali and Roi, 2010).

In this study, we investigated the correlation between the decadal changes in the drivers and flood quantiles; therefore, the long-term evolution of the drivers (quantified by smoothed time-series through a LOESS regression) influences the results rather than their exact value. For this reason, we tested the impact of this choice in two ways:

- 1. We calculate and compare the trend in antecedent precipitation for all catchments using windows of 30, 45 and 60 days (i.e. AP30, AP45 and AP60, see Fig. SC1). Figure SC1 shows that the spatial pattern and magnitude of the trend in AP30 (Fig. SC1a) are very similar to those obtained with longer temporal windows (Fig. SC1b and SC1c).
- 2. We visually compare the long-term evolution of AP30, AP45 and AP60 in a number of catchments randomly selected in northwestern, southern and eastern Europe (regions defined as in Bertola et al., 2020) for different ranges of catchment area (see Fig. SC2). Small (<100 km2), medium (100-1000 km2), large (1000-10000 km2) and very large catchments (>10000 km2) are compared. Figure SC2 shows that, despite the value of AP increases with longer temporal windows (as expected), its long-term oscillation is nearly the same (in the range 30-60 days). This is observed for catchment sizes and regions.

We conclude that using a longer temporal window for antecedent precipitation would not significantly change the results of our study, even for very large catchments.

We prefer not to use the antecedent precipitation index (API) because additional assumptions on its parameters would be required (i.e. the decay factor and maximum lag parameter, as defined in Woldemeskel and Sharma, 2016). Furthermore, the maximum lag parameter (similar to the temporal window of the antecedent precipitation) is often assumed as 14 days or less (e.g. Mediero et al., 2014; Woldemeskel and Sharma, 2016). Even though we prefer not to change the soil moisture index, we will mention the issue in the discussion of the revised paper.

We would like to thank Yves Tramblay for spotting the error in the reference; we will correct it in the revised manuscript.

Figure SC1: trend of catchment-averaged antecedent precipitation for temporal windows of 30 (a), 45 (b) and 60 (c) days for each station over the period 1960-2010.

---

## Referee Comment (RC1) · Anonymous Referee #1 · 23 Sep 2020

The manuscript *Do small and large floods have the same drivers of change? A regional attribution analysis in Europe* by Bertola et al is the natural sequel of the previous HESS paper by some of the same authors (https://doi.org/10.5194/hess-24-1805-2020) taking the investigation from the detection to the attribution of changes in high flows of different frequencies. The manuscript is well organised and deals with a very interesting topic which I imagine will attract many readers. It is highly relevant for a European readership and presents an investigation of which physical variables appear to drive the magnitude of high flows in Europe differentiation between the common and the extreme high flows.

In the introduction the authors frame their study within the current literature giving a nice excursus of what the current state of modelling change is. I have some disagreement on some of the language they use, though. They mention several papers saying that most studies focus on the change in the mean annual flood, which they then contrast to their interesting new approach. On the other hand though most studies I have seen in the literature (including those cited) focus on explaining the change in the location parameter (or sometimes the scale parameter) - but typically the mean flood would be a combination of all distribution parameters. So modelling a change in location typically reflect on a change in the mean flood, but the model aims at modelling some slightly different quantity. More importantly, when location and scale are both allowed to change the mean flood would change as a function of both parameters, so the model for the mean flood would be rather complex.

In equation 4 it is not very clear to me how the model is regional and each station contribute information to the model. I understand that all station-years contribute to the likelihood and things are then corrected using the likelihood inflation? I mean this is not a multilevel model in which station-specific parameters are allowed, is that right?

Further, I understand that the model for the two quantities is estimated at the same time, so the $q_2$ is "hidden" in the $x_{100}$ model: to make this maybe more obvious I would use a bracket before to "connect" equation 4a and 4b.

I am also not entirely sure why no $\epsilon_g$ was allowed in the growth factor model. For those who might want to code this up themselves it might be helpful to have the formulae translating parameters to quantile and even more, to be able to read the Stan code - I would recommend that the authors share their code either via GitHub or via some more academic-oriented repository such as Zenodo.

To summarise: I think the model could be described with more details, especially for those who have not read the first paper on which this builds.

Finally this is more of a curiosity, I was wandering what forms do the parameters functions take when one re-transforms the quantiles back to parameters. Can these shapes tell us something interesting about what types of functional relationship exist between the physical variables and the distribution parameters?

I find the modelling strategy of the authors quite interesting because they effectively model two quantiles which are indeed of interest rather than the parameters: should we then ditch the standard parametrisation of the Gumbel distribution or are the parameters still useful?

Regarding the choice of the priors: the authors choose to set a hard bound on the elasticity parameters: did this create any problem in the estimation? I mean: is the posterior distribution very concentrated on the lower bound or does it spread nicely?

I am somewhat dubious about the pooling of stations done by the authors and the use of averaged quantities across the rather large 200km x 200 km squares. To begin with the pooling will necessarily pool together information on small basins and large basins: this might not be problematic but I am more worried that with such large squares the pooling will put together very different types of basins (for example, alpine small basins and lowland larger basins): the response these basins have to drivers might be very different. Since from my understanding there aren't station specific parameters in the model, there might be some issues with the homogeneity of the groups and the ability of the model to identify the effect of the drivers on high flows. On the other hand, the average value of such large square might be not very useful to explain the variability of high flows for small basins and possibly inflate the variability of the results. I don't really see a way of out of this - I think the authors made some pragmatic decisions to be able to perform their study, but I wander whether we can fully trust their findings. In a similar

vein: some areas are much more densely gauged than others, allowing possibly for a more precise estimation. This is not mentioned at all in the current manuscript.

Figure 8 is very interesting, but maybe I would complement it with two other visuals which would be relevant: the changes in the precipitation, soil moisture and snowmelt in each of the regions (to make more sense of how the curves morph from row to row in Figure 8) and final change in the different quantiles between the beginning and the end of the recording period (or any two moments in time).

---

## Author Comment (AC3) · 12 Oct 2020

**"Do small and large floods have the same drivers of change? A regional attribution analysis in Europe"**

by M. Bertola, A. Viglione, S. Vorogushyn, D. Lun, B. Merz and G. Blöschl

We reproduce and number in the following document all the comments of the Referee *in italic characters*, followed by our answers.

**Anonymous Referee #1**

*The manuscript "Do small and large floods have the same drivers of change? A regional attribution analysis in Europe" by Bertola et al is the natural sequel of the previous HESS paper by some of the same authors (https://doi.org/10.5194/hess-24-1805-2020) taking the investigation from the detection to the attribution of changes in high flows of different frequencies. The manuscript is well organised and deals with a very interesting topic which I imagine will attract many readers. It is highly relevant for a European readership and presents an investigation of which physical variables appear to drive the magnitude of high flows in Europe differentiation between the common and the extreme high flows.*

> We thank the Anonymous Referee #1 for the time she/he spent on our manuscript and for the useful and constructive comments that will help to improve the quality of the manuscript. We have carefully considered and addressed all her/his comments in the following.

*1. In the introduction the authors frame their study within the current literature giving a nice excursus of what the current state of modelling change is. I have some disagreement on some of the language they use, though. They mention several papers saying that most studies focus on the change in the mean annual flood, which they then contrast to their interesting new approach. On the other hand though most studies I have seen in the literature (including those cited) focus on explaining the change in the location parameter (or sometimes the scale parameter) - but typically the mean flood would be a combination of all distribution parameters. So modelling a change in location typically reflect on a change in the mean flood, but the model aims at modelling some slightly different quantity. More importantly, when location and scale are both allowed to change the mean flood would change as a function of both parameters, so the model for the mean flood would be rather complex.*

> In the introduction we meant that most of the studies about flood changes in the literature focus on changes in mean flood behaviour (not necessarily the mean annual flood), and they do not explicitly account for changes in flood quantiles with large return periods. For example, at lines 27-29 we refer to the trend detection studies that use tests (e.g. the Mann-Kendall test or the Petitt test) to detect changes in the mean flood behaviour.
>
> As the referee says, most studies in the literature about non-stationary frequency analyses allow the location parameter (and, more rarely, the scale parameter) to vary

in time using time-varying covariates, as described in the manuscript at lines 92-97. This translates to changes in the mean annual flood (or in flood quantiles), which is a function of the distribution parameters, although changes in model parameters are modelled. In our approach we focus on flood quantiles, rather than distribution parameters, in order to explicitly model the relationships between small and large floods (i.e. $q_2$ and $q_{100}$) and the potential drivers of flood change, and to separate the effects of drivers on floods associated with different return periods.

We will clarify this terminology in the introduction of the revised manuscript.

*2. In equation 4 it is not very clear to me how the model is regional and each station contribute information to the model. I understand that all station-years contribute to the likelihood and things are then corrected using the likelihood inflation? I mean this is not a multilevel model in which station-specific parameters are allowed, is that right?*

The referee is right; all station-years contribute to the likelihood and the likelihood is corrected using the magnitude adjustment to account for spatial cross-correlation between sites. This is not a multilevel model and the only station-specific parameter is the error term $\varepsilon_g$ which accounts for the additional local variability not explained by catchment area and the covariates. We will clarify it in the revised manuscript.

*3. Further, I understand that the model for the two quantities is estimated at the same time, so the q2 is "hidden" in the x100 model: to make this maybe more obvious I would use a bracket before to "connect" equation 4a and 4b.*

It will be done in the revised manuscript.

*4. I am also not entirely sure why no $\varepsilon_g$ was allowed in the growth factor model. For those who might want to code this up themselves it might be helpful to have the formulae translating parameters to quantile and even more, to be able to read the Stan code - I would recommend that the authors share their code either via GitHub or via some more academic-oriented repository such as Zenodo.*

The error term is not included in the growth factor $x'_{100}$ because we make the assumption that the growth curve is the same across all sites within the region, while the median flood is allowed to vary between sites. This is similar to the index flood method of Dalrymple (1960) and Hosking and Wallis (1997). We will better explain it in the revised manuscript. In sect 2.1 of the revised manuscript we will also add the relationships linking the Gumbel parameters $\xi$ and $\sigma$ to $q_2$ and $x'_{100}$ (as in Bertola et al., 2020), i.e.:

$$q_2 = \xi + \sigma \cdot y_2$$
$$x'_{100} = \sigma(y_{100} - y_2)/(\xi + \sigma \cdot y_2) \qquad (A1)$$

where $y_2 = -\ln(-\ln(0.5))$ and $(y_{100} - y_2) = -\ln(-\ln(0.99)) + \ln(-\ln(0.5))$.

We will share the stan code in GitHub with the revised manuscript.

*5. To summarise: I think the model could be described with more details, especially for those who have not read the first paper on which this builds.*

We thank the referee for her/his suggestions, we will improve the description of the model as detailed by our answers to the specific questions above (see answers to points 2 to 4).

*6. Finally this is more of a curiosity, I was wandering what forms do the parameters functions take when one re-transforms the quantiles back to parameters. Can these shapes tell us something interesting about what types of functional relationship exist between the physical variables and the distribution parameters?*

The relationships linking $q_2$ and $x'_{100}$ to the Gumbel parameters can be obtained inverting eq. A1:

$$\xi = q_2 \left(1 - x'_{100} \frac{y_2}{y_{100} - y_2}\right)$$
$$\sigma = \frac{q_2 x'_{100}}{y_{100} - y_2}$$

(A2)

The regional change model adopted is:

$$\ln q_2 = \ln \alpha_{2_0} + \gamma_{2_0} \ln S + \alpha_{2_1} \ln X_1 + \alpha_{2_2} \ln X_2 + \alpha_{2_3} \ln X_3 + \varepsilon$$
$$\ln x'_{100} = \ln \alpha_{g_0} + \gamma_{g_0} \ln S + \alpha_{g_1} \ln X_1 + \alpha_g \ln X_2 + \alpha_{g_3} \ln X_3$$

(A3)

By substituting Eq. A3 into Eq. A2, we get:

$$\xi = \exp\left(\ln \alpha_{2_0} + \gamma_{2_0} \ln S + \sum_{i=1}^{3} \alpha_{2_i} \ln X_i + \varepsilon\right) - \left(\frac{y_2}{y_{100} - y_2}\right) \cdot \exp(\ln \alpha_{2_0} + \ln \alpha_{g_0} +$$
$$(\gamma_{2_0} + \gamma_{g_0}) \ln S + \sum_{i=1}^{3} (\alpha_{2_i} + \alpha_{g_i}) \ln X_i + \varepsilon)$$

$$\sigma = \frac{1}{y_{100} - y_2} \cdot \exp(\ln \alpha_{2_0} + \ln \alpha_{g_0} + (\gamma_{2_0} + \gamma_{g_0}) \ln S + \sum_{i=1}^{3} (\alpha_{2_i} + \alpha_{g_i}) \ln X_i + \varepsilon)$$

As a result, the functional relationships between the drivers and Gumbel parameters have a similar structure to those expressed in Eq. A3 between the alternative parameters and the drivers. Additionally, we observe that the second term in the location parameter has little influence, given that $\frac{y_2}{y_{100} - y_2} \cong 0.086$.

*7. I find the modelling strategy of the authors quite interesting because they effectively model two quantiles which are indeed of interest rather than the parameters: should we then ditch the standard parametrisation of the Gumbel distribution or are the parameters still useful?*

From a practical perspective, flood quantiles are clearly attractive, since flood risk managers are indeed interested into these quantities (e.g. the 100-year flood) and their changes in time. In this study, we directly model the changes in flood quantiles because, in a Bayesian framework it is typically easier for experts to formulate prior beliefs in terms of flood quantiles associated with large return periods, which they are familiar with, rather than in terms of distribution parameters (see, e.g., the causal information expansion based on expert judgement in Viglione et al., 2013). The distribution parameters are nevertheless fundamental as they determine the location and spread of the distribution.

*8. Regarding the choice of the priors: the authors choose to set a hard bound on the elasticity parameters: did this create any problem in the estimation? I mean: is the posterior distribution very concentrated on the lower bound or does it spread nicely?*

The introduction of these hard bounds in the priors is done in order to hydrologically 'inform' the attribution analysis. The elasticity parameters are, in fact, reasonably expected to be positive, given the selected drivers (corresponding to changes of the same sign in drivers and floods). In most of the cases/regions that we considered across Europe, the posterior distributions of the model parameters look nicely spread.

The referee is kindly referred to the example figure below (Fig. A1), where the posterior distribution of the elasticity of $q_2$ and $q_{100}$ to the three drivers are shown for the three regions analysed in Sec. 3.3, located in northwestern, southern and eastern Europe (see Fig. 1 of the manuscript for the location of these three regions), respectively. In few cases, when the covariate change and the flood change have different signs, the posterior distribution of the related elasticity parameter is concentrated on the lower bound. This can be observed for example in Fig. A1c in the case of the posterior distribution of the elasticity of $q_T$ to antecedent precipitation, which slightly increases over time, while flood magnitude decreases for both T=2 and 100 years. We will clarify it in the revised manuscript

[Figure]

Figure A1: Posterior distribution of the elasticity of $q_T$ to the drivers in three regions respectively located in northwestern (a), southern (b) and eastern (c) Europe.

*9. I am somewhat dubious about the pooling of stations done by the authors and the use of averaged quantities across the rather large 200km x 200 km squares. To begin with the pooling will necessarily pool together information on small basins and large basins: this might not be problematic but I am more worried that with such large squares the pooling will put together very different types of basins (for example, alpine small basins and lowland larger basins): the response these basins have to drivers might be very different. Since from my understanding there aren't station specific parameters in the model, there might be some issues with the homogeneity of the groups and the ability of the model to identify the effect of the drivers on high flows. On the other hand, the average value of such large square might be not very useful to explain the variability of high flows for small basins and possibly inflate the variability of the results. I don't really see a way of out of this - I think the authors made some pragmatic decisions to be able to perform their study, but I wander whether we can fully trust their findings. In a similar vein: some areas are much more densely gauged than others, allowing possibly for a more precise estimation. This is not mentioned at all in the current manuscript.*

In this study we are interested in the average regional behaviour and flood attribution at the large scale. The results of the study should therefore be interpreted at the continental scale as average contributions of the drivers to flood changes in the regions, rather than at the catchment scale.

As in Bertola et al. (2020), flood data of multiple sites are pooled in this study within spatial windows of size of 600km×600km, with an overlapping length of 200km in both directions. The size and overlapping length of the windows were chosen in Bertola et

al. (2020) after several preliminary tests, in order to ensure a sufficient number of gauges within each window and an appropriate spatial resolution at which to present the regional trends at the continental scale. Significant differences in spatial change pattern were not observed when changing the window size. The rationale behind the homogeneity assumption is that the spatial windows are characterized by comparatively homogeneous climatic conditions, flood generation processes and processes driving flood changes. The attribution analysis is thereby performed at the regional scale, where average regional contributions of the decadal changes in the drivers to average regional trends in flood quantiles are estimated. We have not assessed the statistical homogeneity of the regions in terms of the flood change model used here. One reason is that formal procedures to assess the regional homogeneity, such as those used in regional flood frequency analysis (e.g. Hosking and Wallis, 1993; Viglione et al., 2007), are not available in the context of the present model. Also, while deviation from regional homogeneity would probably invalidate estimates of local flood change statistics from the regional information (e.g. in the prediction in ungauged basins; see Blöschl et al., 2013), we expect its effect on the average regional behaviour to be less relevant.

We will acknowledge and clarify this assumption in the revised manuscript.

Catchment area (S) and the drivers (X1, X2, X3) are indeed station specific. The average regional values shown in Fig. 4-7 are obtained for hypothetical catchment area of 1000 $km^2$ and for average changes in the drivers in each region over the period 1960-2010.

The different density of stations across Europe clearly influences the precision of the estimation and it is taken into account by the width of the credible bounds, represented for each region by white circles in Fig. 4-7. We will mention this in the revised manuscript.

*10. Figure 8 is very interesting, but maybe I would complement it with two other visuals which would be relevant: the changes in the precipitation, soil moisture and snowmelt in each of the regions (to make more sense of how the curves morph from row to row in Figure 8) and final change in the different quantiles between the beginning and the end of the recording period (or any two moments in time).*

We thank the referee for her/his suggestion. We will add this information to Figure 8.

**References:**

Bertola, M., Viglione, A., Lun, D., Hall, J., and Blöschl, G.: Flood trends in Europe: are changes in small and big floods different?, Hydrology and Earth System Sciences, 24, 1805–1822, https://doi.org/10.5194/hess-24-1805-2020, 2020.

Blöschl, G., Sivapalan, M., Wagener, T., Viglione, A., and Savenije, H.: Runoff Prediction in Ungauged Basins: Synthesis across Processes, Places and Scales, Cambridge University Press, https://doi.org/10.1017/CBO9781139235761, 2013.

Dalrymple, T.: Flood frequency methods, US Geological Survey, water supply paper A, 1543, 11–51, 1960.

Hosking, J. R. M. and Wallis, J. R.: Some statistics useful in regional frequency analysis, Water Resour. Res., 29, 271–281, https://doi.org/10.1029/92WR01980, 1993.

Hosking, J. R. M. and Wallis, J. R.: Regional Frequency Anal- ysis, Cambridge University Press, Cambridge, UK, p. 240, ISBN 0521430453, 1997.

Viglione, A., Laio, F., and Claps, P.: A comparison of homogeneity tests for regional frequency analysis, Water Resour. Res., 43, 1– 10, https://doi.org/10.1029/2006WR005095, 2007.

Viglione, A. *et al.* (2013) 'Flood frequency hydrology: 3. A Bayesian analysis', *Water Resources Research*, 49(2), pp. 675–692. doi: 10.1029/2011WR010782.

---

## Referee Comment (RC2) · Anonymous Referee #2 · 24 Oct 2020

The manuscript addresses an important and popular topic in statistical hydrology: how and why are floods changing? General: The paper is well written and rooted in the literature. Flood attribution in this study is limited to three drivers, and the focus is on which physical process is relatively more important (as compared with other studies which have attempted to attribute floods to many different processes, e.g. Schlef et al 2019). I think the paper presents an interesting analysis, though I think its main conclusions and results are not well interpreted for the general scientific community, limiting the applicability and generalizability of the work. I give specific details in this review. The manuscript's results do not convince me of the conclusions - maybe it is in the presentation - but I am not really convinced that the conclusions about the

observed flood changes are valid. Without sufficient validation of the approach, its unclear if the method performed as expected. For example, in choosing out a case from the dataset, can we validate that in fact within a region, extreme precip increased and floods increased for a q2 or q100 return period (not from spatial difference / relative contribution plots, but from actual time series and data within that region?

Introduction: While I agree in general that focusing on the mean/median can mask changes in the various return periods of a flow distribution, the mean is also traditionally an indicator of changes within the distribution and thus is an important piece of the story on how nonstationarity may be impacting a particular basin. Also after reading the paper, I am not certainly convinced that extreme precipitation is well aligned with the 100-yr event and would like to see more on the bounds of the 2 and 100 yr return periods. Speaking of return periods: in the spirit of helping to change the conversation from return periods to a more meaningful statistic, like reliability, I would recommend reframing the need to examine changes in floods from 'return period based' to something more robust. At the very least, return period must be well defined at the start of this paper: When the authors refer to return period in the manuscript, I think they mean "average return period" (e.g. Read and Vogel, 2015). Additional issues with the use of return period here: Please describe how the formulation of 2.1 holds true when p = 1/T is no longer valid..). Can the Gumbel parameters be inferred from the 2- and 100-yr floods if the distribution is changing? I do not follow why the method for extreme precipitation was used. I am assuming there is a reason that this was made more complicated than pairing the flood data with the rainfall data in a more straightforward way. In using the average occurrence day, is there a chance that the actual highest precip/flood days are left out of the analysis (for example if they do not occur within the average window)?

Lines 270-272: With regard to elasticities specifically, why was a decadal % used to identify drivers? In the results generally, the interpretations of the individual elements are limited. For example, 273 "Extreme precipitation contributes positively to flood

changes in northwestern and central Europe, and negatively in southern and eastern Europe". Also 276-77: "The contributions of snowmelt to changes in q2 and q100 are predominantly negative and marked in Eastern Europe, with small differences towards smaller contributions in absolute values with return period". Its a bit of work for the reader to translate this, using Fig 5. Put this in terms that are clearly translatable. This issue persists throughout the results, and clarification could especially be helpful in regard to the elasticities discussion. Translate elasticity into the meaningful metric that it is (e.g. in lines 286-88, an elasticity of zero indicates XXX).

Again, in 4.2, the results are restated, but are not communicated in a way that is useful for an audience. Take it a step further to explain how the sign/magnitude of the changes/relative contributions are meaningful in the context of the problem this manuscript is addressing. The Conclusions (4.4) really offer no further insight for the reader than the results. Suggest revising this to focus on implications.

Minor Comments: Fig 8: why use a hypothetical catchment instead of select from those available within the study region? lines 345, 348 spell out numbers below ten.

---

## Author Response (AR1)

**"Do small and large floods have the same drivers of change? A regional attribution analysis in Europe"**

by M. Bertola, A. Viglione, S. Vorogushyn, D. Lun, B. Merz and G. Blöschl

We would like to thank the Editor and the two Anonymous Reviewers for the time they spent on our manuscript and for the positive and constructive comments.

For the sake of convenience, we reproduce and number in the following document all the comments of the Reviewers in *italic characters*, followed by our answers. Together with the revised manuscript in PDF we also send the Marked Manuscript in PDF in which all the changes in the text are tracked (deleted in  characters, while new text is in blue characters). Numbers in brackets (highlighted in yellow) indicate the line numbers in the Marked Manuscript.
* * *
**Editor**

*Editor Decision: Reconsider after major revisions (further review by editor and referees)*

*Comments to the Author:*

*Dear Authors,*

*Thank you for your prompt and detailed responses to the reviews. I find this is a very interesting and original manuscript which is likely to be of considerable interest to the flood hydrology community, however there are some legitimate queries about whether the method performs as expected. Both reviewers have raised some valuable points (especially better clarifying and validating the methodology, and interpreting the results). I would therefore like to invite you to please submit a revised manuscript which addresses the issues raised by the reviewers. The manuscript will then be sent to the same reviewers for re-review.*

*I am looking forward to reading your revised manuscript.*

*Regards,*
*Louise Slater*

We thank the Editor Louise Slater for this chance to improve our manuscript. We have carefully considered and addressed all the comments provided by the Reviewers, as detailed in the following pages.
* * *
**Anonymous Referee #1**

*The manuscript "Do small and large floods have the same drivers of change? A regional attribution analysis in Europe" by Bertola et al is the natural sequel of the previous HESS paper by some of the same authors (https://doi.org/10.5194/hess-24-1805-2020) taking the investigation from the detection to the attribution of changes in high flows of different frequencies. The manuscript is well organised and deals with a very interesting topic which I imagine will attract many readers. It is highly relevant for a European readership and presents an investigation of which physical variables appear to drive the magnitude of high flows in Europe differentiation between the common and the extreme high flows.*

We thank the Anonymous Referee #1 for the time she/he spent on our manuscript and for the useful and constructive comments that will help to improve the quality of the manuscript. We have carefully considered and addressed all her/his comments in the following.

*1. In the introduction the authors frame their study within the current literature giving a nice excursus of what the current state of modelling change is. I have some disagreement on some of the language they use, though. They mention several papers saying that most studies focus on the change in the mean annual flood, which they then contrast to their interesting new approach. On the other hand though most studies I have seen in the literature (including those cited) focus on explaining the change in the location parameter (or sometimes the scale parameter) - but typically the mean flood would be a combination of all distribution parameters. So modelling a change in location typically reflect on a change in the mean flood, but the model aims at modelling some slightly different quantity. More importantly, when location and scale are both allowed to change the mean flood would change as a function of both parameters, so the model for the mean flood would be rather complex.*

In the introduction we meant that most of the studies about flood changes in the literature focus on changes in mean flood behaviour (not necessarily the statistical mean of maximum annual flood discharges), and they do not explicitly account for changes in flood quantiles with large return periods. For example, at lines 28-29 we refer to the trend detection studies that use tests (e.g. the Mann-Kendall test or the Petitt test) to detect changes in the mean flood behaviour.

As the referee rightly says, most studies in the literature about non-stationary frequency analyses allow the location parameter (and, more rarely, the scale parameter) to vary in time using time-varying covariates, as described in the revised manuscript at lines 94-99. This translates to changes in the mean annual flood (or in flood quantiles), which is a function of the distribution parameters, although changes in model parameters are modelled. In our approach we have reparametrised the Gumbel model to use flood quantiles as distribution parameters, in order to explicitly model the relationships between small and large floods (i.e. $q_2$ and $q_{100}$) and the potential drivers of flood change, and to separate the effects of drivers on floods associated with different return periods. We have clarified this terminology in the abstract (lines 1-3) and in the introduction of the revised manuscript (lines 94, 100-104).

*2. In equation 4 it is not very clear to me how the model is regional and each station contribute information to the model. I understand that all station-years contribute to the likelihood and things are then corrected using the likelihood inflation? I mean this is not a multilevel model in which station-specific parameters are allowed, is that right?*

> The referee is right; all station-years contribute to the likelihood and the likelihood is corrected using the magnitude adjustment to account for spatial cross-correlation between sites. This is not a hierarchical model and the only station-specific term is the error term $\varepsilon$ which accounts for the additional local variability not explained by catchment area and the covariates. We have clarified it in the revised manuscript in section 2.1 (lines 140-143) and section 2.6 (lines 267-269).

*3. Further, I understand that the model for the two quantities is estimated at the same time, so the q2 is "hidden" in the x100 model: to make this maybe more obvious I would use a bracket before to "connect" equation 4a and 4b.*

> It has been done in the revised manuscript (Eq. 5a and 5b).

*4. I am also not entirely sure why no $\varepsilon_g$ was allowed in the growth factor model. For those who might want to code this up themselves it might be helpful to have the formulae translating parameters to quantile and even more, to be able to read the Stan code - I would recommend that the authors share their code either via GitHub or via some more academic-oriented repository such as Zenodo.*

> The error term is not included in the growth factor $x'_{100}$ because we make the assumption that the growth curve is the same across all sites within the region, while the median flood is allowed to vary between sites. This is similar to the index flood method of Dalrymple (1960) and Hosking and Wallis (1997). We have better explained it in the revised manuscript (lines 143-146).

> In sect 2.1 of the revised manuscript we have also added the relationships linking the Gumbel parameters $\xi$ and $\sigma$ to $q_2$ and $x'_{100}$ (Eq. 2a and 2b), as in Bertola et al. (2020).

> We have shared the Stan code via GitHub and the link is provided in the 'Code availability' section of the revised manuscript.

*5. To summarise: I think the model could be described with more details, especially for those who have not read the first paper on which this builds.*

> We thank the referee for her/his suggestions, we have improved the description of the model as detailed by our answers to the specific questions above (see answers to points 2 to 4).

*6. Finally this is more of a curiosity, I was wandering what forms do the parameters functions take when one re-transforms the quantiles back to parameters. Can these shapes tell us something interesting about what types of functional relationship exist between the physical variables and the distribution parameters?*

> The relationships linking $q_2$ and $x'_{100}$ to the Gumbel parameters can be obtained inverting eq. 2a,b:

$$\xi = q_2 \left(1 - x'_{100} \frac{y_2}{y_{100} - y_2}\right)$$

$$\sigma = \frac{q_2 x'_{100}}{y_{100} - y_2}$$

$$(A1)$$

The regional change model adopted is:

$$lnq_2 = ln\alpha_{2_0} + \gamma_{2_0}lnS + \alpha_{2_1}lnX_1 + \alpha_{2_2}lnX_2 + \alpha_{2_3}lnX_3 + \varepsilon$$
$$lnx'_{100} = ln\alpha_{g_0} + \gamma_{g_0}lnS + \alpha_{g_1}lnX_1 + \alpha_g lnX_2 + \alpha_{g_3}lnX_3$$

$$(A2)$$

By substituting Eq. A2 into Eq. A1, we get:

$$\xi = exp\left(ln\alpha_{2_0} + \gamma_{2_0}lnS + \sum_{i=1}^{3}\alpha_{2_i}lnX_i + \varepsilon\right) - \left(\frac{y_2}{y_{100}-y_2}\right) \cdot exp\left(ln\alpha_{2_0} + ln\alpha_{g_0} + \right.$$
$$\left. (\gamma_{2_0} + \gamma_{g_0})lnS + \sum_{i=1}^{3}(\alpha_{2_i} + \alpha_{g_i})lnX_i + \varepsilon\right)$$

$$\sigma = \frac{1}{y_{100}-y_2} \cdot exp\left(ln\alpha_{2_0} + ln\alpha_{g_0} + (\gamma_{2_0} + \gamma_{g_0})lnS + \sum_{i=1}^{3}(\alpha_{2_i} + \alpha_{g_i})lnX_i + \varepsilon\right)$$

As a result, the functional relationships between the drivers and Gumbel parameters have a similar structure to those expressed in Eq. A2 between the alternative parameters and the drivers. Additionally, we observe that the second term in the location parameter has little influence, given that $\frac{y_2}{y_{100}-y_2} \cong 0.086$.

*7. I find the modelling strategy of the authors quite interesting because they effectively model two quantiles which are indeed of interest rather than the parameters: should we then ditch the standard parametrisation of the Gumbel distribution or are the parameters still useful?*

From a practical perspective, flood quantiles are clearly attractive, since flood risk managers are indeed interested in these quantities (e.g. the 100-year flood) and their changes in time. In this study, we directly model the changes in flood quantiles because, in a Bayesian framework it is typically easier for experts to formulate prior beliefs in terms of flood quantiles associated with large return periods, which they are familiar with, rather than in terms of distribution parameters (see, e.g., the causal information expansion based on expert judgement in Viglione et al., 2013). The distribution parameters are nevertheless fundamental as they independently determine the location and spread of the distribution. We have added this consideration in the discussion Section (lines 421-423).

*8. Regarding the choice of the priors: the authors choose to set a hard bound on the elasticity parameters: did this create any problem in the estimation? I mean: is the posterior distribution very concentrated on the lower bound or does it spread nicely?*

The introduction of these hard bounds in the priors is done in order to hydrologically 'inform' the attribution analysis. The elasticity parameters are, in fact, reasonably expected to be positive, given the selected drivers (corresponding to changes of the same sign in drivers and floods). In most of the cases/regions that we considered across Europe, the posterior distributions of the model parameters look nicely spread. The referee is kindly referred to the example figure below (Fig. A1), where the posterior distribution of the elasticity of $q_2$ and $q_{100}$ to the three drivers are shown for the three regions analysed in Sec. 3.3, located in northwestern, southern and eastern

Europe (see Fig. 1 of the manuscript for the location of these three regions), respectively. In few cases, when the covariate change and the flood change have different signs, the posterior distribution of the related elasticity parameter is concentrated on the lower bound. This can be observed for example in Fig. A1c in the case of the posterior distribution of the elasticity of $q_T$ to antecedent precipitation, which slightly increases over time, while flood magnitude decreases for both T=2 and 100 years. Similar considerations can be drawn from Fig. 8 and the newly added Fig. 9 of the manuscript, showing respectively the posterior median and 90% credible bounds of the elasticities and the average changes in time of the drivers and flood quantiles in the regions. We have clarified it in Sect 2.5 (lines 247-252) and in Sect 3.3 (lines 368-375) of the revised manuscript.

[Figure]

Figure A1: Posterior distribution of the elasticity of $q_T$ to the drivers in three regions respectively located in northwestern (a), southern (b) and eastern (c) Europe.

*9. I am somewhat dubious about the pooling of stations done by the authors and the use of averaged quantities across the rather large 200km x 200 km squares. To begin with the pooling will necessarily pool together information on small basins and large basins: this might not be problematic but I am more worried that with such large squares the pooling will put together very different types of basins (for example, alpine small basins and lowland larger basins): the response these basins have to drivers might be very different. Since from my understanding there aren't station specific parameters in the model, there might be some issues with the homogeneity of the groups and the ability of the model to identify the effect of the drivers on high flows. On the other hand, the average value of such large square might be not very useful to explain the variability of high flows for small basins and possibly inflate the variability of the results. I don't really see a way of out of this - I think the authors made some pragmatic decisions to be able to perform their study, but I wander whether we can fully trust their findings. In a similar vein: some areas are much more densely gauged than others, allowing possibly for a more precise estimation. This is not mentioned at all in the current manuscript.*

In this study we are interested in the average regional behaviour and flood attribution at the large scale. The results of the study should therefore be interpreted at the continental scale as average contributions of the drivers to flood changes in the regions, rather than at the catchment scale.

As in Bertola et al. (2020), flood data of multiple sites are pooled in this study within spatial windows of size of 600km×600km, with an overlapping length of 200km in both

directions. The size and overlapping length of the windows were chosen in Bertola et al. (2020) after several preliminary tests, in order to ensure a sufficient number of gauges within each window and an appropriate spatial resolution at which to present the regional trends at the continental scale. Significant differences in spatial change pattern were not observed when changing the window size. The rationale behind the homogeneity assumption is that the spatial windows are characterized by comparatively homogeneous climatic conditions, flood generation processes and processes driving flood changes. The attribution analysis is thereby performed at the regional scale, where average regional contributions of the decadal changes in the drivers to average regional trends in flood quantiles are estimated. We have not assessed the statistical homogeneity of the regions in terms of the flood change model used here. One reason is that formal procedures to assess the regional homogeneity, such as those used in regional flood frequency analysis (e.g. Hosking and Wallis, 1993; Viglione et al., 2007), are not available in the context of the present model. Also, while deviation from regional homogeneity would probably invalidate estimates of local flood change statistics from the regional information (e.g. in the prediction in ungauged basins; see Blöschl et al., 2013), we expect its effect on the average regional behaviour to be less relevant. We have acknowledged and clarified this assumption in Sect. 2.6 (lines 269-270) and Sect 4.3 (lines 434-448) of the revised manuscript.

Catchment area (S) and the drivers (X1, X2, X3) are indeed station specific. The average regional values shown in Fig. 4-7 are obtained for hypothetical catchment area of 1000 km$^2$ and for average changes in the drivers in each region over the period 1960-2010.

The different density of stations across Europe clearly influences the precision of the estimation and it is taken into account by the width of the credible bounds, represented for each region by white circles in Fig. 4-7. We have mentioned this in the revised manuscript (lines 301-304).

*10. Figure 8 is very interesting, but maybe I would complement it with two other visuals which would be relevant: the changes in the precipitation, soil moisture and snowmelt in each of the regions (to make more sense of how the curves morph from row to row in Figure 8) and final change in the different quantiles between the beginning and the end of the recording period (or any two moments in time).*

We thank the referee for her/his suggestion. We have introduced one additional figure (Fig. 9), where flood and driver time series are shown for each of the three regions analysed in Sect. 3.3, as well as their average changes in time within the regions (numbers in the panels). Additional text related to Fig. 9 has also been added in Sect. 3.3 to complement the results of Fig. 8 (lines 354-356, 363, 372-375).
* * *
**Anonymous Referee #2**

*The manuscript addresses an important and popular topic in statistical hydrology: how and why are floods changing? General: The paper is well written and rooted in the literature. Flood attribution in this study is limited to three drivers, and the focus is on which physical process*

*is relatively more important (as compared with other studies which have attempted to attribute floods to many different processes, e.g. Schlef et al 2019). I think the paper presents an interesting analysis, though I think its main conclusions and results are not well interpreted for the general scientific community, limiting the applicability and generalizability of the work. I give specific details in this review.*

We thank the Anonymous Referee #2 for the time she/he spent on our manuscript and for the useful and constructive comments that will help to improve the quality of the manuscript. We have carefully considered and addressed all her/his comments in the following.

For the sake of clarity, the choice of the three drivers was driven by the results of recent studies (i.e. Blöschl et al. 2017, 2019; Berghuijs et al., 2019, Kemter at al. 2020) that pointed out potential correlations between timing and magnitude of floods and extreme precipitation, soil moisture and snowmelt, across Europe (it has been clarified in lines 412-414 of the revised manusript). This study aims at formally quantifying the contribution of these drivers to flood changes, i.e. 'flood change attribution' and not 'flood attribution', as done in Schlef et al. (2019).

*1. The manuscript's results do not convince me of the conclusions - maybe it is in the presentation - but I am not really convinced that the conclusions about the observed flood changes are valid. Without sufficient validation of the approach, it's unclear if the method performed as expected. For example, in choosing out a case from the dataset, can we validate that in fact within a region, extreme precip increased and floods increased for a q2 or q100 return period (not from spatial difference / relative contribution plots, but from actual time series and data within that region?*

We thank the reviewer for this comment; we understand that our approach should be further clarified in the manuscript.

In the revised manuscript we have shown actual time series and average flood and driver changes in one additional figure (Fig. 9), to support our results for the three example regions analyzed in Sect. 3.3 (in line with the request of the anonymous reviewer #1, see reply to SC1, nr. 10). Based on this additional figure we can indeed demonstrate that, within a region (e.g., Northwestern Europe), extreme precipitation increased, and floods increased for a $q_2$ and a $q_{100}$ return period. Additional text related to Fig. 9 has been added in Sect. 3.3 to complement the results of Fig. 8 (lines 354-356, 363, 372-375).

This study more generally suggests that the changes in flood quantiles potentially caused by the three considered drivers are overall compatible, in terms of patterns and magnitude, with the flood changes observed in previous studies (e.g. Blöschl et al., 2019; Bertola et al., 2020). Some discrepancies are nevertheless observed, for instance, in Scandinavia, where the contributions of the drivers are all positive or close to zero, while mostly moderate negative flood trends were observed in previous studies (see Sect. 4.2). In Sect. 4.2 we commented on possible reasons for this discrepancy (e.g. other potential drivers not accounted for in this study). We have clarified this in the revised manuscript (lines 404-406).

On the other hand, we did not cross-validate the model against data from additional stations or for other periods of time, because we do not aim at estimating driver

contributions locally, in ungauged basins, nor at extrapolating the results of the model to the future. We are instead interested in the average driver contributions to changes in flood quantiles over the five analysed decades, and the results should be interpreted at the European scale. Additionally, in order to avoid spurious correlations and to make sure that hydrologically meaningful contributions are identified, in the Bayesian framework we adopted informative prior distributions of the elasticity parameters (i.e. the parameters controlling the relationship between flood and driver changes), based on expert judgement and qualitative reasoning (Sect. 2.5). In practice, the informative prior distributions reflect the fact that flood and driver changes are expected to have the same sign (e.g. floods increased because precipitation increased, and positive flood changes cannot be attributed to negative precipitation changes). We have clarified this in the revised manuscript (lines 248-252 and lines 446-448).

*2. Introduction: While I agree in general that focusing on the mean/median can mask changes in the various return periods of a flow distribution, the mean is also traditionally an indicator of changes within the distribution and thus is an important piece of the story on how nonstationarity may be impacting a particular basin. Also after reading the paper, I am not certainly convinced that extreme precipitation is well aligned with the 100-yr event and would like to see more on the bounds of the 2 and 100 yr return periods. Speaking of return periods: in the spirit of helping to change the conversation from return periods to a more meaningful statistic, like reliability, I would recommend reframing the need to examine changes in floods from 'return period based' to something more robust. At the very least, return period must be well defined at the start of this paper: When the authors refer to return period in the manuscript, I think they mean "average return period" (e.g. Read and Vogel, 2015). Additional issues with the use of return period here: Please describe how the formulation of 2.1 holds true when p = 1/T is no longer valid..). Can the Gumbel parameters be inferred from the 2- and 100-yr floods if the distribution is changing? I do not follow why the method for extreme precipitation was used. I am assuming there is a reason that this was made more complicated than pairing the flood data with the rainfall data in a more straightforward way. In using the average occurrence day, is there a chance that the actual highest precip/flood days are left out of the analysis (for example if they do not occur within the average window)?*

Introduction: We agree with the reviewer. We have clarified this point in the introduction of the revised manuscript (lines 30-31).

Return periods: In this study, we analyze changes in time of selected flood quantiles q, which are associated with fixed annual exceedance probability 1-p (in the notation used in the manuscript) through the quantile function q(p, $\xi(t), \sigma(t)$). In a non-stationary context, the pdf is a function of time and, consequently, also the flood quantiles (associated with fixed annual exceedance probabilities) change with time. The Gumbel parameters can be inferred from (time dependent) flood quantiles, associated with fixed exceedance probabilities. In the manuscript we refer to the return periods, rather than the annual exceedance probabilities, because they are widely used and understood in the engineering practice. Therefore, for ease of interpretation, the return period T is obtained from the annual exceedance probability 1-p through the relationship p=1-1/T, although other formulations are available under non-stationarity conditions. We do refer to the average return period as defined, for example, in Read and Vogel (2015). In this study, we directly model the changes in flood quantiles because, in a Bayesian framework it is typically easier for experts to

formulate prior beliefs in terms of flood quantiles, which they are familiar with, rather than in terms of distribution parameters (see, e.g., the causal information expansion based on expert judgement in Viglione et al., 2013). This has been mentioned in the discussion section (lines 421-423). Examples of return period terminology used in a similar non-stationary context in the literature are Renard et al. (2006), Machado et al. (2015), Šraj et al. (2016). For these reasons we prefer to maintain the return period terminology in the manuscript. However, we have clarified the terminology used in the introduction (lines 102-105) and in the method section 2.1 (lines 128-129). We prefer not to use the reliability (as defined in Read and Vogel, 2015) instead of the return period in this context because it requires the additional definition of the lifetime of a system/project. However, in the revised manuscript we have mentioned the existence of alternative ways of communicating event likelihood in stationary and non-stationary contexts, such as the reliability (line 105).

Extreme precipitation: We did not pair floods with the corresponding event precipitation because we do not aim at doing event attribution, but at attributing flood changes to the long-term evolution of the drivers in the average season of occurrence of floods. In other words, we use flood seasonality to identify drivers that are typically relevant for the generation of the annual peaks. The variability of flood seasonality in each station is taken into account by the width of the time window that is used to extract the 7-day maximum precipitation and snowmelt (i.e. if floods occur evenly distributed throughout the year, the width of the window is 12 months, and if floods occur always on the same date, this window is reduced to 3 months). We have clarified it in Sect. 2.4 of the revised manuscript (lines 197-200).

*3. Lines 270-272: With regard to elasticities specifically, why was a decadal % used to identify drivers? In the results generally, the interpretations of the individual elements are limited. For example, 273 "Extreme precipitation contributes positively to flood changes in northwestern and central Europe, and negatively in southern and eastern Europe". Also 276-77: "The contributions of snowmelt to changes in q2 and q100 are predominantly negative and marked in Eastern Europe, with small differences towards smaller contributions in absolute values with return period". It's a bit of work for the reader to translate this, using Fig 5. Put this in terms that are clearly translatable. This issue persists throughout the results, and clarification could especially be helpful in regard to the elasticities discussion. Translate elasticity into the meaningful metric that it is (e.g. in lines 286-88, an elasticity of zero indicates XXX).*

The elasticities (Fig. 4, lines 298-315) are measured in %/% and represent the change in flood quantiles due to 1% change in one driver. Lines 317-329 and Fig. 5 refer to the contributions of the drivers to changes in q2 and q100. These contributions are obtained by multiplying the elasticity of flood quantiles to the drivers by the average trend of the corresponding driver over the region (in %/decade). The contributions are measured in %/decade because they represent the fraction of the trend in flood quantiles (which is measured in %/decade) that is explained by a specific driver. We have clarified this and rephrased the sentences pointed out by the reviewer through Sect. 3.2 and 3.3.

*4. Again, in 4.2, the results are restated, but are not communicated in a way that is useful for an audience. Take it a step further to explain how the sign/magnitude of the changes/relative contributions are meaningful in the context of the problem this manuscript is addressing. The Conclusions (4.4) really offer no further insight for the reader than the results. Suggest revising this to focus on implications.*

We have rephrased section 4.2 and revised section 4.4 in the revised manuscript according to the suggestions of the reviewer.

*Minor Comments:*

*5. Fig 8: why use a hypothetical catchment instead of selecting from those available within the study region?*

The aim of this study is to estimate the average contributions of drivers to changes in flood quantiles within each region, and not for a single catchment in the region. For this reason, we do not represent the results for a catchment area corresponding to one specific existing catchment in the region. Instead, we show the average regional driver contributions for one hypothetical medium-sized catchment (i.e. catchment area of 1000 km$^2$). Consequently, the results should be read and interpreted at the large continental scale, rather than at the local (i.e. catchment) scale. We have clarified this in Sect. 2.6 of the revised manuscript (lines 278-282).

*6. lines 345, 348 spell out numbers below ten.*

They have been spelled out in the revised manuscript

**References:**

Berghuijs, W. R. *et al.* (2019) 'The Relative Importance of Different Flood-Generating Mechanisms Across Europe', *Water Resources Research*, pp. 1–12. doi: 10.1029/2019WR024841.

Bertola, M. *et al.* (2020) 'Flood trends in Europe: are changes in small and big floods different?', *Hydrology and Earth System Sciences*, 24(4), pp. 1805–1822. doi: 10.5194/hess-24-1805-2020.

Blöschl, G., Sivapalan, M., Wagener, T., Viglione, A., and Savenije, H. (2013), 'Runoff Prediction in Ungauged Basins: Synthesis across Processes, Places and Scales', *Cambridge University Press*, https://doi.org/10.1017/CBO9781139235761.

Blöschl, G. *et al.* (2017) 'Changing climate shifts timing of European floods', *Science*, 357(6351), pp. 588–590. doi: 10.1126/science.aan2506.

Blöschl, G. *et al.* (2019) 'Changing climate both increases and decreases European river floods', *Nature*, 573(7772), pp. 108–111. doi: 10.1038/s41586-019-1495-6.

Dalrymple, T. (1960), 'Flood frequency methods', US Geological Survey, water supply paper A, 1543, 11–51.

Hosking, J. R. M. and Wallis, J. R. (1993), 'Some statistics useful in regional frequency analysis', *Water Resour. Res.,* 29, 271–281, https://doi.org/10.1029/92WR01980.

Hosking, J. R. M. and Wallis, J. R. (1997), 'Regional Frequency Analysis', *Cambridge University Press*, Cambridge, UK, p. 240, ISBN 0521430453.

Kemter, M., Merz, B., Marwan, N., Vorogushyn, S., and Blöschl, G.: Joint Trends in Flood Magnitudes and Spatial Extents Across Europe, Geophysical Research Letters, 47, 1–8, https://doi.org/10.1029/2020GL087464, 2020.

Machado, M. J. *et al.* (2015) 'Flood frequency analysis of historical flood data under stationary and non-stationary modelling', *Hydrology and Earth System Sciences*, 19(6), pp. 2561–2576. doi: 10.5194/hess-19-2561-2015.

Renard, B., Lang, M. and Bois, P. (2006) 'Statistical analysis of extreme events in a non-stationary context via a Bayesian framework: case study with peak-over-threshold data', *Stochastic Environmental Research and Risk Assessment*, 21(2), pp. 97–112. doi: 10.1007/s00477-006-0047-4.

Schlef, K.E., Moradkhani, H. & Lall, U. (2019) 'Atmospheric Circulation Patterns Associated with Extreme United States Floods Identified via Machine Learning', *Sci Rep* **9,** 7171 https://doi.org/10.1038/s41598-019-43496-w

Šraj, M. *et al.* (2016) 'The influence of non-stationarity in extreme hydrological events on flood frequency estimation', *Journal of Hydrology and Hydromechanics*, 64(4), pp. 426–437. doi: 10.1515/johh-2016-0032.

Viglione, A., Laio, F., and Claps, P. (2007) 'A comparison of homogeneity tests for regional frequency analysis', *Water Resources Research*, 43, 1– 10, https://doi.org/10.1029/2006WR005095.

Viglione, A. *et al.* (2013) 'Flood frequency hydrology: 3. A Bayesian analysis', *Water Resources Research*, 49(2), pp. 675–692. doi: 10.1029/2011WR010782.